# Towards Fully FP8 GEMM LLM Training at Scale

**Alejandro Hernández-Cano**[*]
EPFL
alejandro.hernandezcano@epfl.ch

**Dhia Garbaya**[*]
EPFL
dhia.garbaya@epfl.ch

**Imanol Schlag**
ETHZ
ischlag@ethz.ch

**Martin Jaggi**
EPFL
martin.jaggi@epfl.ch

## Abstract

Despite the significant potential of FP8 data formats for large language model (LLM) pre-training, their adoption has been limited due to challenges in maintaining stability at scale. Existing approaches often rely on suboptimal fine-grained FP8 kernels or fall back to higher-precision matrix multiplications (GEMMs) in sensitive components, such as attention projections, compromising potential throughput gains. We introduce a new class of LLM architectures that, for the first time, support FP8 computation for all GEMMs within transformer blocks during both forward and backward passes. This enables unprecedented throughput gains, particularly at scale, while matching the downstream performance of standard BF16 training. Our architecture design reduces large outlier activations, promoting stable long-term FP8 training. In addition, we identify key metrics to monitor low-precision training and predict potential future divergences.

## 1   Introduction

Recent progress in the training of transformer-based Large Language Models (LLMs) has significantly advanced the field of language modelling. Scaling up both model size and training data remains a reliable strategy to enhance their performance [15]. Consequently, state-of-the-art models are typically trained at scale using extensive datasets [1, 7, 25], requiring substantial computational resources—often in the order of millions of GPU hours.

Thus, the development of efficient training techniques has become increasingly essential. One of the main research avenues for efficiency is the use of lower-precision number formats to accelerate training on appropriate hardware accelerators. Recently, the use of 8-bit floating-point (FP8) formats has shown promising results [3, 6, 24]. However, the widespread adoption of current approaches is still limited due to suboptimal throughput benefits. One cause of slowdowns is the use of higher precision in those General Matrix Multiplications (GEMMs) which are most sensitive, such as attention score computation, while another issue is the overhead caused by more granular FP8 scaling strategies. One of the key challenges in 8-bit LLM training originates from the relatively narrow dynamic range offered by FP8 formats and thus higher risk of underflows and overflows, especially with the prevalence of large outlier features observed in the LLM's neural activations during training [4, 30, 9]. We formalize the effect of outliers on quantization later in Appendix C. To mitigate this issue, modern FP8 training recipes utilise various scaling techniques before casting from higher-precision formats—typically BF16 [14] for activations—to FP8 formats used in matrix multiplications. These scaling approaches help maximize the effective use of FP8's limited dynamic range, reducing the risk of underflows and overflows.

---

[*]Equal contribution.

39th Conference on Neural Information Processing Systems (NeurIPS 2025).

Recent work has introduced promising FP8 training recipes by employing multiple scaling factors per single tensor [3], allowing for a finer and more precise casting to lower precision. Yet, this comes with an efficiency overhead, diminishing the large gains initially expected from using FP8. Another strategy involves adjusting the standard SwiGLU-based transformer architecture [28] to prevent emergent outliers from occurring [6]. This area of optimization remains underexplored, as most works focus on FP8 GEMMs within the linear projections within the transformer, while maintaining higher precision for other GEMMs, namely those involved in the dot product attention mechanism. *We refer to such training strategies simply as* FP8 *training. We label the approach of also including FP8 attention computation as* FP8DPA.

In this paper, we introduce **FOG**: the **F**ast and **O**utlier-**G**uarded set of LLM architectures specifically designed to mitigate large activation outliers and enable efficient large-scale FP8 training with low-overhead scaling strategies. For the first time to our knowledge, this approach enables FP8 GEMMs not only in the linear projection, but also within the attention mechanism of each transformer block, achieving unprecedented throughput improvements of up to 43% in the 8B parameter model scale, while maintaining equivalent downstream performance compared to higher precision baselines. In addition, we present a comprehensive recipe for monitoring, explaining, and predicting training instabilities that might not surface in the early stages of training. This approach provides researchers with greater confidence in the long-term stability of FP8 training recipes, reducing the need for costly, full-scale experiments when using new architectures. Furthermore, we provide an interestingly useful observation about larger models' tendency to diverge later in training with FP8.

Our key contributions are the following:

- We introduce the FOG set of architectures, designed to minimise outlier features during training. Our recipe allows stable training with FP8 computation of all GEMMs inside the transformer blocks, surpassing the throughput of the standard BF16 approach by up to **43%**.

- Our design achieves equivalent quality results to BF16 baselines, while providing a significant speed-up. We empirically attest both performance and stability on various model sizes (0.4B, 1.5B, 8B) and data regimes up to 15x the Chinchilla optimal data budget [12].

- We show the flexibility of FOG design as it can be adapted to several architectures including various families of activation functions and even Mixture-of-Experts (MoE) settings.

- Using kurtosis, we provide a recipe to judge an architecture's robustness to FP8 training in long data regimes using diagnostics from shorter runs. We use this recipe to explain previously observed divergence behaviour at scale, and offer a wide range of empirical results to demonstrate its usefulness. We believe this contribution allows FP8 training insights on future transformer variants developed by the community, without the need for expensive full-scale experiments.

## 2 Background

Due to its limited dynamic range, FP8 tensors are particularly prone to overflows and underflows when representing extreme values. The FP8 formats come in two standard forms [20]: E4M3 and E5M2, each with different trade-offs. The first format, with four exponent bits and three mantissa bits, offers higher precision. In contrast, the E5M2 format, with five exponent bits and two mantissa bits, provides a broader dynamic range at the cost of reduced precision. Existing large-scale distributed training frameworks such as DeepSpeed or Megatron [29] leverage this distinction by employing E4M3 for tensors in the forward pass to maintain precision and E5M2 for the backward pass to handle the broader dynamic range of gradients effectively. Nonetheless, both formats have much lower representation capacity than half- or single-precision formats. Therefore, various scaling strategies are applied when casting tensors down to FP8 in order to make more efficient use of this restricted range. These strategies are mainly tensorwise and fall into two main categories: delayed scaling and just-in-time scaling (JIT). Delayed scaling uses information from previous training iterations to determine the scaling factor of the tensor for the ongoing iteration, requiring a single pass on the data along with storing a short history of useful metrics observed across an interval of past iterations. JIT scaling, on the other hand, can hinder the gains from using FP8 because it uses the distribution of the tensor being produced—in higher precision—to compute the scalar, before casting the input and performing the GEMM in FP8, requiring at least two passes through data. A more recent approach aims to make scaling more robust by using multiple scaling factors per tensor, allowing different

tensor blocks to have different scaling factors [27, 3]. This leads to a more precise FP8 casting within each block. Naturally, this finer scaling strategy induces a larger overhead on such GEMM kernels relative to the tensorwise delayed scaling recipe.

Ensuring stable FP8 training remains challenging. It becomes problematic when certain activations produce large outliers during training, making such a low-precision representation unfeasible and leading to rapid divergence. Prior work introduced the term *massive activations*, a phenomenon similar to outlier features, and showed their crucial role in LLMs' capabilities [30]. Understanding the dynamics of these outliers is crucial for explaining FP8 divergence and identifying the network components responsible for them. One notable source of such outliers' amplification has been identified to be the widely adopted SwiGLU (Swish Gated-Linear-Unit) activation function [28]. Replacing it with a scaled variant, SmoothSwiGLU regulates large outliers and was shown to stabilize previously diverging FP8 training runs and ensure their convergence [6].

Further examination has shown that not only is SwiGLU an outlier amplifier, but Gated Linear Units (GLUs) in general, as well as pre-normalization layers, suggesting that improper signal propagation is the root cause of outliers [9]. Removing these components and equipping transformers with QK entropy regularization mechanisms such as QK RMS Normalization [10], producing the Outlier Protected (OP) architectures [9], has been shown to diminish late-stage outliers observed by orders of magnitude, while providing equivalent prediction quality. While OP architectures were shown to be beneficial for post-training quantization, its use for FP8 pre-training remains unexplored. Finally, an alternative to pre-normalization layers are post-normalization layers [18]. Long data regime trainings have confirmed their superiority in terms of training stability with the standard BF16 mixed precision training [21].

# 3  FOG: Fast and Outlier-Guarded FP8-suited architectures

Our architecture base, as illustrated in Figure 1, makes key changes to widely-used transformer networks [32]. The pre-normalization block before the attention mechanism and FFN is removed. In addition, a normalization mechanism in the attention is added to prevent entropy collapse, a key training instability in transformers [34], from occurring. This mechanism can take the form of a QK RMSNorm block [10]:

$$\mathcal{N}_{\boldsymbol{\gamma}}(\boldsymbol{x}) := \frac{1}{\mathrm{rms}(\mathbf{x})} \boldsymbol{\gamma} \odot \mathbf{x}, \qquad \mathrm{rms}(\mathbf{x}) := \frac{\|\mathbf{x}\|_2}{\sqrt{D}},$$

where $\mathbf{x} \in \mathbb{R}^D$, $\boldsymbol{\gamma} \in \mathbb{R}^N$ is the learnable *gains* vector, $\odot$ is the Hadamard product and $\| \cdot \|_2$ is the $\ell_2$-norm. Alternatively, the $\tanh_\alpha(x) := \tanh(\alpha x)$ element-wise activation function, where $\alpha \in \mathbb{R}$ is trainable, can be applied to query and key tensors. This activation has been shown to have regularization effects akin to RMS normalization blocks [35], while being computationally more efficient.

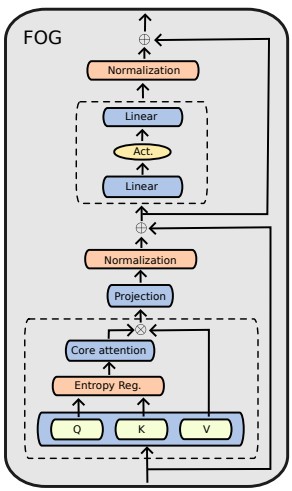

Figure 1: **FOG transformer.**

Further, the input of the first transformer block is scaled by $\sigma^{-1}$ to maintain unit variance activations at initialization, where $\sigma$ is the chosen standard deviation of the network's random initialization. Finally, to enhance performance, a learnable normalization block is applied before the residual connections. This takes the form of a LayerScale [31] block, $\mathrm{LayerScale}_{\boldsymbol{\gamma}}(\mathbf{x}) := \boldsymbol{\gamma} \odot \mathbf{x}$, where $\boldsymbol{\gamma} \in \mathbb{R}^D$ is a learnable gain vector, or an RMSNorm block, resulting in a post-normalized architecture [18]. In both cases, the learnable gains vector is initialized to $1/\sqrt{\mathrm{num\_layers}}$ and keeps the residual branch unnormalized, allowing proper signal propagation [9]. Our architecture suite is specified in Table 1, and further details are available in Appendix B.

While the OP architecture already offers several guards to prevent large outliers from occurring, we observed that it remains an impractical choice for FP8DPA training. In Section 5.1 we show that, like all other architectures tested, it suffers a fatal loss divergence early during training. We isolate the two components responsible for OP's incompatibility with FP8DPA training: the trainable QK RMSNorm gains vector $\boldsymbol{\gamma}$, and the lack of any normalization. We identify the use of post-normalization as not prone to the outlier tendency pre-normalization networks have.

| Model | QK-Regularization | Activation | Normalization |
|---|---|---|---|
| FOG-max | RMSNorm* | xIELU [13] | Post-RMSNorm |
| FOG-opt | RMSNorm* | GeLU | Post-RMSNorm |
| FOG-flash | Tanh* | GeLU | Post-RMSNorm |
| OP [9] | RMSNorm | GeLU | LayerScale |

Table 1: **FOG architecture suite compared with OP.** Regularizations marked with * indicate that gains are not trainable. Each variant offers different trade-offs, with FOG-flash having the higher throughput and FOG-max observed to have better downstream quality.

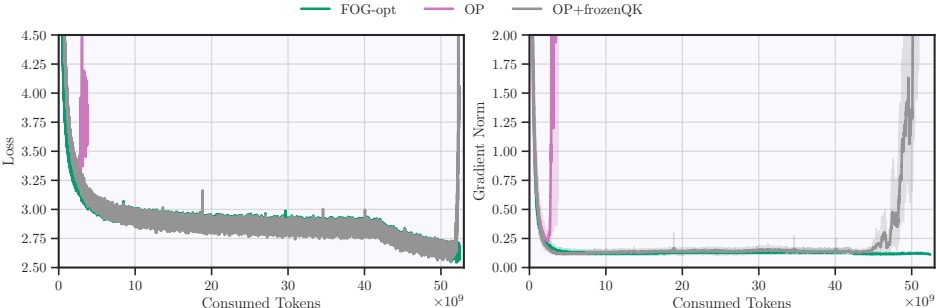

Figure 2: **From OP to FOG-opt step by step.** Comparison of 390M models under FP8DPA training. The first architecture to diverge is OP, while OP with frozen QK RMSNorm gains survives the stable phase of training. It still, however, experiences a significant divergence during the learning rate cooldown, which starts around 42B tokens in. The architecture that converges, FOG-opt, is the result of adding post-normalization to the previous recipe. Gradient norm reported is the 200-rolling-window mean and 5%-95% quantile bands.

Figure 2 ablates the components transitioning from OP to FOG-opt. We can see that freezing the trainable QK RMSNorm gains results in a significantly more stable training. We attribute the early divergence of OP to the fact that uncontrolled QK normalization leads to an explosion of its gains when training in low precision. Note that these gains are generally not weight-decayed. We experimentally observe this explosion, confirm that using $L_2$ regularization helps delay the divergence. We finally opted for freezing the gains to a constant value as it is simpler and sufficient, doesn't compromise performance, and offers a small speedup. Our ablations highlight that a constant value for the gains slightly greater than 1 improves loss. Therefore, to retain its benefit after removing the $\gamma$ gains vector, we increase the standard $s = 1/\sqrt{D_{qk}}$ attention softmax scale–a tiny optimization trick offering equivalent attention score matrix $\mathbf{S}$:

$$\mathbf{S} = \frac{1}{\sqrt{D_{qk}}} \mathcal{N}_{\gamma_0}(\mathbf{Q}) \mathcal{N}_{\gamma_0}(\mathbf{K})^{\top} = \frac{1}{\sqrt{D_{qk}}} \left( \frac{\gamma_0 \mathbf{Q}}{\mathrm{rms}(\mathbf{Q})} \right) \left( \frac{\gamma_0 \mathbf{K}}{\mathrm{rms}(\mathbf{K})} \right)^{\top} = \frac{\gamma_0^2}{\sqrt{D_{qk}}} \mathcal{N}_{\mathbf{1}}(\mathbf{Q}) \mathcal{N}_{\mathbf{1}}(\mathbf{K})^{\top},$$

Finally, we empirically show that the addition of post-normalization is important to ensure convergence with FP8DPA during the learning rate decay phase.

Prior works also favored post-normalization over pre-normalization [21], providing evidence of their better stability in BF16 training. We extend this observation to our FP8 setting and we confirm that learnable LayerScale blocks alone, even with controlled QK regularization, cannot ensure convergence during this last phase. We attribute this late divergence of OP to the fact that LayerScale blocks without normalizations are not enough to handle FP8 outliers, potentially due to the considerable changes in model statistics following the learning rate decay, that are summed up in the residual connections resulting in huge activation outliers for last layers, as highlighted by the increasing pattern of outliers on each transformer block's output in Figure 3. We note that we initially tested the idea of cooling down the previously constant weight decay during the learning cooldown phase, aiming to conserve model weights' norm [16]. The Appendix G shows that such intervention has no noticeable effect on stability neither performance. We chose to conserve this decision for all FOG runs for consistency and fair comparison across ablations.

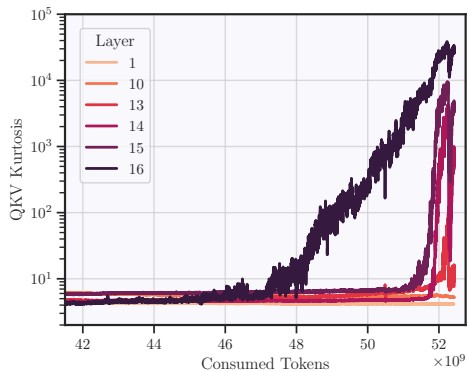

Figure 3: **Kurtosis of QKV tensors during FP8DPA learning rate cooldown with OP+frozenQK architecture.** Later layers exhibit significantly larger activation outliers.

## 4 Long-term outlier dynamics

To analyse the outliers present in neural network activations, we use kurtosis as a metric of the extremity of deviations of activation values (such as by outliers). We define the kurtosis $\mathrm{kurt}(\mathbf{x})$ of a vector $\mathbf{x} \in \mathbb{R}^D$ as the scalar

$$\mathrm{kurt}(\mathbf{x}) := \frac{\mu[\mathbf{x}^4]}{\sigma^2[\mathbf{x}^2]},$$

where $\mu$ and $\sigma^2$ are the sample mean and variance, respectively, and exponentiation is taken element-wise. Given an activation tensor $\mathbf{X} \in \mathbb{R}^{N \times C \times D}$, where $N$, $C$, and $D$ are the batch size, sequence length, and hidden size respectively, we define its kurtosis as the average $\mathrm{kurt}(\mathbf{X}) := \frac{1}{NC} \sum_{n=1}^{N} \sum_{c=1}^{C} \mathrm{kurt}(\mathbf{x}_{nc})$.

Under this definition, $\mathrm{kurt}(\mathbf{x})$ is maximized when few elements of $\mathbf{x}$ reach extremely large values, relative to the variance across the entire vector, i.e., when large outlier features are present. This definition has been used to analyse outliers in BF16 training in previous work [9] and, unlike the standard definition of kurtosis [22] in the probability theory literature, this definition does not center $\mathbf{x}$ to have zero-mean. For our use, this is consistent with the fact that FP8 kernels do not shift their inputs before scaling and casting down. We track the dynamics of kurtosis in key activations. Namely, the inputs of the second projection in FFNs, the QKV matrix, and the output of each transformer block. Unless explicitly stated, we report the average activation kurtosis across all layers.

Using these activations, we can analyse the emergence of large outlier features at different stages during training. Figure 4 demonstrates an equivalent loss progression to the baseline while offering up to orders of magnitude lower kurtosis in some activations. Note that, unlike previous FP8 approaches, FOG architectures are trained with FP8 attention computations, introducing more quantization errors. As a result, the kurtosis of key, query, and value projections becomes particularly relevant.

**Activation Functions** Baseline Llama exhibits late divergence during FP8 training (with attention in BF16), which has been attributed solely to the quadratic behavior of its gated activation function—emerging when weights become sufficiently aligned late in training [6]. In our extended 450B token run using the FOG-max architecture, we employ the inherently quadratic xIELU activation function, see Equation (2), and observe stable training with kurtosis levels orders of magnitude lower than those of baseline Llama. In fact, modifying the FOG-max architecture to use the **SwiGLU** activation function resulted in stable FP8DPA training behaviour, as disscussed in Appendix D. These results strongly suggests that architectures biased towards low kurtosis activations during training enable the stable use of quadratic activations, and challenges the completeness of prior explanations. This is particularly interesting given that such activations are known to produce linear gradients, which benefit the backward pass—likely contributing to FOG-max's superior performance over GeLU-based variants as seen in Section 5.3.

**Long-term outlier growth** These architectures exhibit a sub-linear to logarithmic trend in the long-term growth of QKV outliers, as consistently shown by kurtosis in Figure 4. This behavior

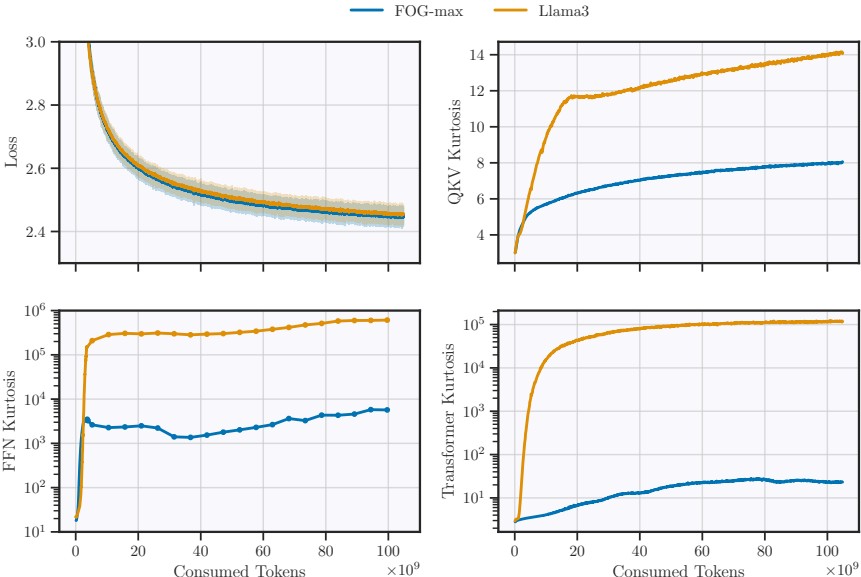

Figure 4: **Loss and kurtosis training dynamics** of 1.5B FOG-max and Llama3 models trained for over 100B tokens with BF16 precision. Loss reported is the 200-rolling-window mean and 5%-95% quantile bands.

supports their robustness to FP8DPA, as it suggests that prohibitively longer training would be required to see a substantial increase in kurtosis. Our extended run is consistent with the hypothesis as it does not exhibit any sign of divergence.

**Kurtosis early signal** Figure 5 shows an example of a diverging FP8DPA run, comparing it with the successful FOG-max training. This emphasises the importance of tracking tensor-level metrics such as kurtosis to potentially predict later divergences, before common global metrics like the loss and gradient norms show any symptoms of divergence. In this example, while loss irrefutably diverged around the 15B token mark and the gradient norm consistently spiked no earlier than 12B tokens, the QKV kurtosis was already diverging from the expected sub-linear growth consistently seen across different architectures as early as the 3B mark, giving a potential early divergence sign.

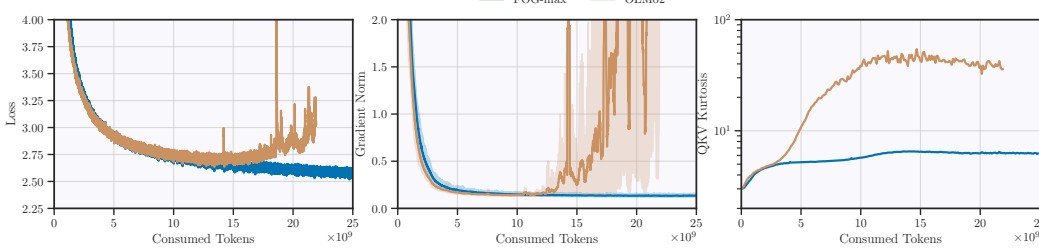

Figure 5: **Training dynamics of a failed and a successful FP8DPA run.** Kurtosis exhibits atypical behaviour much earlier than when the loss diverged. Gradient norm reported is the 200-rolling-window mean and 5%-95% quantile bands.

## 5   Experimental Results

We perform extensive experiments to verify our architecture across several scales. We use the FineWeb-Edu [23] text corpus, filtering out any web opt-out domains with robots.txt, resulting in a rigorous data-compliant corpus [5]. The data is tokenised using a 131K vocabulary BPE

tokenizer. We keep a consistent context length of 4096 during all main experiments. In terms of the optimisation algorithm, we use AdamW [19] with default hyperparameters. Our learning rate schedule is comprised of three phases: Warm-up, Steady, and Decay phases (WSD), as it has been shown to provide equivalent performance to the cosine schedule [8], while allowing to train beyond fixed training durations. For the models, we train 390M, 1.5B and 8B parameter models for different token counts, specified at each experiment. Our baseline architecture follows the Llama3 8B model design [7], with the 390M and 1.5B being adapted to their respective sizes. Since Llama3 uses a gated linear unit, unlike the OP and FOG variants, we increase the FFN sizes of OP and FOG to maintain an equal parameter count. Further details regarding architectures and hyperparameters are available in Appendices A and B.

Our hardware infrastructure consists of nodes with 4 Nvidia Grace Hopper GPUs each. Our distributed training framework is adapted from Megatron-LM [29], which uses Transformer Engine [2] FP8 recipes. With 390M parameters, our experiments reach 50B tokens. We scaled 1.5B experiments to 125B tokens to obtain more meaningful evaluations. In addition to the absence of late-in-training outlier amplification from FOG's non-gated activation functions and our kurtosis progression guarantees, we further validate our method's stability on long data regimes by continuing pretraining FOG-max up to 450B tokens. Finally, we scale the model size to 8B and train for 20B tokens. We show the divergence of other architectures with FP8DPA while FOG variants converge and match the baseline Llama3 BF16 loss, while being 35-43% faster. During all experiments, we use the FP8 delayed scaling strategy, with a margin of zero and a history length of 1024 steps.

We make our implementation, along with detailed steps for our experiments, public under the repository `https://github.com/anonymous4375934/FOG`.

## 5.1 FP8 stability

We compare our approach with different architectures proposed in the literature. Namely, the OP architecture, OLMo2, Llama3, and Llama3 with the SmoothSwiGLU activation following the previous work [6], adapting each network to 390M and 1.5B parameter count. In the case of the Llama3 baseline, we also provide results on the 8B scale. Results are shown in Figure 6. This experiment displays the unsuitability of existing architectures for FP8DPA training, as all of them diverge. For the case of the OP and OLMo2 architecture, despite having an attention outlier-mitigation strategy—the QK RMSNorm—divergence is still observed, as discussed in Section 3.

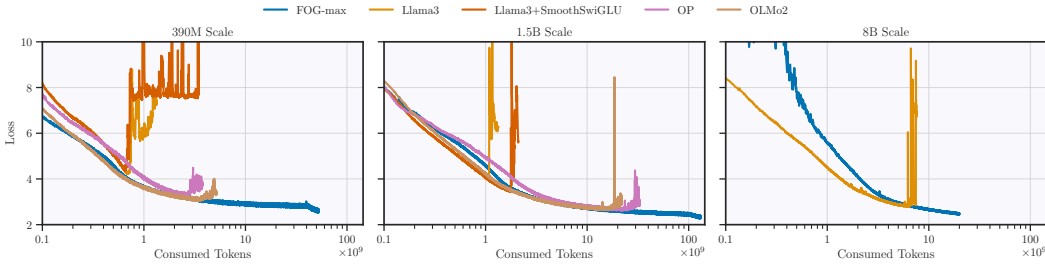

Figure 6: **Cross-entropy loss plots of different architectures with FP8DPA training.** No other tested architecture was able to surpass the 20B token mark without diverging at any scale.

Another interesting observation from these experiments is the tendency of larger models to diverge in later stages of training compared with similar but smaller models. We validate its consistency across architectures, as presented in Table 2. This observation has not been raised before, possibly due to the longer time needed for FP8 settings (with BF16 attention) to diverge, in contrast to FP8DPA training. While this trend could have many practical implications, exploring it fully falls outside the scope of this work, and we encourage future research in this direction.

| Architecture | Model Size | Divergence Mark (in billions of tokens) |
|---|---|---|
| Llama3 | 390M | 0.7 |
| Llama3 | 1.5B | 1.1 |
| Llama3 | 8B | 6.6 |
| OLMo2 | 390M | 3.3 |
| OLMo2 | 1.5B | 15.9 |

Table 2: **Token mark when loss was observed to diverge.**

## 5.2 Efficiency

**Standard context length**   Table 3 explores the efficiency of FOG at 1.5B and 8B model scales under FP8DPA training, using a standard context length of 4096. We compare our set-up with the BF16 baseline and the stable Llama FP8 training with SmoothSwiGLU, which is, to the best of our knowledge, the only dense architecture proposal demonstrated to work at scale with FP8. Note that the SmoothSwiGLU cannot benefit from enabling FP8 GEMMs in the attention mechanism, as it was shown to suffer a big loss divergence in Figure 6. Note the increase in throughput gains as the model size increases. The GEMM input tensors increase in size and consume significantly more time during the overall forward-backward pass, compared with other operators.

| Size | Model | Precision | Throughput (tokens/second/GPU) |
|---|---|---|---|
| 8B | Llama | BF16 | 9105 |
| | Llama+SmoothSwiGLU | FP8 | 12228 (+34.3%) |
| | FOG-max | FP8DPA | 12344 (+35.5%) |
| | FOG-opt | FP8DPA | 12414 (+36.3%) |
| | FOG-flash | FP8DPA | **12764 (+40.2%)** |
| 1.5B | Llama | BF16 | 46470 |
| | FOG-max | FP8DPA | 53551 (+15.2%) |
| | FOG-opt | FP8DPA | 53877 (+15.9%) |
| | FOG-flash | FP8DPA | 54848 (+18.0%) |
| | Llama+SmoothSwiGLU | FP8 | **54903 (+18.1%)** |

Table 3: **Training throughput measures with FOG versus other baselines.** Using eight GH200 nodes with Zero-1 sharding [26] for 8B models and a single GH200 node for 1.5B models. Notably, in the 8B scale, all FOG variants outperform other architectures.

**Long-context scenario**   Enabling stable training under FP8DPA regime leads to great benefits in long context scenarios, because the throughput becomes bottlenecked by the dot product attention computation with quadratic complexity. As a result, we observe larger speed-up gaps between FOG and prior FP8 approaches as sequence length increases, as demonstrated in Table 4.

| Context | FOG-flash | Llama+SmoothSwiGLU | Speed-up Gap |
|---|---|---|---|
| 4096 (TP=1) | **+42.6%** | +38.5% | **+4.1%** |
| 8192 (TP=1) | **+43.5%** | Out Of Memory | **–** |
| 8192 (TP=2) | **+39.1%** | +34.2% | **+4.9%** |
| 16384 (TP=2) | **+38.8%** | +31.1% | **+7.7%** |

Table 4: **Training throughput gains under varying sequence lengths** (relative to Llama BF16), performed at 8B scale using 8 GH200 nodes with a global batch size of 1024. Increasing to longer contexts required enabling Tensor Parallelism (TP). Raw throughput values reported in Table 11.

## 5.3 Downstream performance

We compare our proposals with the higher-precision Llama3 baseline across a wide range of standard benchmarks to measure their downstream performance. Inference during down-stream evaluation uses BF16 precision. In Table 5, we report some of the most relevant scores along with an average across a larger set of tasks, detailed in the Appendix G. All FOG variants offer comparable downstream performance with the higher precision Llama3 baseline with FOG-max architecture, even outperforming it. The 1.5B models are trained on 125B tokens, whereas smaller models are trained on 50B tokens.

| Model | Hellaswag | ARC | PIQA | Average* |
|---|---|---|---|---|
| Llama 390M | 33.5 \| - | 47.9 \| - | 65.0 \| - | 39.8 \| - |
| FOG-max | 36.5 \| 36.3 | 62.9 \| 62.5 | 68.0 \| 68.2 | **41.2** \| 40.8 |
| FOG-opt | 36.1 \| 35.6 | 61.5 \| 61.3 | 68.0 \| 67.8 | 40.9 \| 40.4 |
| FOG-flash | 35.9 \| 35.2 | 61.5 \| 60.4 | 68.1 \| 68.2 | 40.5 \| 40.3 |
| Llama 1.5B | 43.7 \| - | 71.8 \| - | 72.5 \| - | 46.1 \| - |
| FOG-max | 43.3 \| 43.4 | 71.6 \| 73.0 | 72.6 \| 73.3 | 46.0 \| **47.1** |
| FOG-opt | 43.3 \| 42.7 | 71.3 \| 70.8 | 72.6 \| 72.0 | 45.7 \| 46.0 |
| FOG-flash | 42.8 \| 41.9 | 70.9 \| 69.4 | 72.2 \| 72.0 | 45.7 \| 44.9 |

Table 5: **Performance across various tasks.** For each task and model size, the first score results from the BF16 ablation and the second from the FP8DPA one. The average* is across a larger set of tasks, show in Appendix G.

## 5.4 Long-data regimes

To further justify the viability of FP8DPA long training with FOG, we train a 1.5B FOG-max on 450B tokens, way beyond the previously identified 200B tokens divergence mark of Llama2-7B [6]. Note that our observation of smaller models' tendancy to diverge earlier with FP8DPA, and the long-term outlier analysis in Section 4 further underline the sufficiency of such a training duration.

We also switch to use FP16 optimizer states and BF16 gradients after 130B tokens, saving up memory previously used by full precision states, gradients, and model parameters master copy. We display the learning dynamics of our approach in Figure 7. The language modeling loss exhibits equivalent to better smoothness compared to the corresponding Llama baseline.

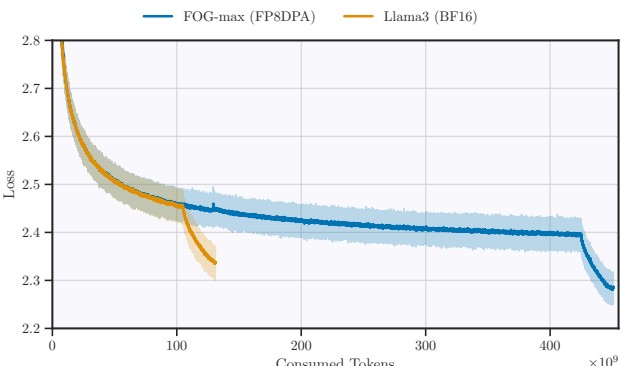

Figure 7: **Long-data training regimes.** FOG-max 1.5B FP8DPA is trained on 450B tokens. The higher precision Llama3 experiment is included as reference. Note that a learning rate cooldown is performed during the last 25B tokens of each experiment, following the WSD schedule. Loss reported is the 200-rolling-window mean and 5%-95% quantile bands.

### 5.5 Additional results

**SwiGLU**   This work mainly studies three architectural variants, sharing in common the use of point-wise activation functions. As mentioned in Section 4, we extended FOG to gated activation functions by showing its stability when using a gated MLP (SwiGLU). Details in Appendix D.

**FP8 optimizer moments**   We ran an additional FP8DPA experiment using FOG-flash following the same 390M scale setup but with 8 bit optimizer moments. The loss converged smoothly to a value of $2.645$, nearly identical to the value $2.649$ obtained with higher-precision moments. More details are available in Appendix E.

**MoE**   We tested FOG under a Mixture-of-Experts (MoE) setting and the training was consistently stable. This experiment further supports the robustness of FOG. More details can be found in Appendix D.

## 6   Limitations

Despite its robustness, record throughput boosts, and flexibility, the FOG set of architectures remains bounded by the following limitation. The final projection (LM head) is still performed in BF16. This operator is known to be very sensitive to outliers and has been used with half-precision in forward-backward FP8 training approaches, including ours. Due to computational constraints, we decided to keep the study of this limitation for future work.

## 7   Conclusion

In this paper we demonstrate, for the first time, stable LLM training with fully FP8 matrix multiplications within the transformer blocks–including the attention mechanism–without sacrificing performance. We tested FP8DPA training across a wide set of previously proposed architectures and show that they consistently diverge early during training, highlighting the difficulty of FP8DPA training and novelty in our results. Moreover, in contrast with other granular scaling recipes, we use the low-overhead delayed scaling FP8 strategy. Our design provides on-par downstream quality with the higher precision baseline, while offering up to **43%** faster training at 8B scale. We scale our 1.5B model to 450B tokens, 15x the Chinchilla-optimal data budget for its size. Our work brings the community one step closer to fully FP8 GEMM training at scale i.e including the language modeling head. We further justify the long-term stability of our architecture by observing the outlier training dynamics across key activations by using kurtosis. The use of kurtosis to track outliers present during training was shown to provide meaningful insights to favour certain architectural components or to predict future instabilities, as it is a quantitative metric that measures outliers.

## Acknowledgement

This work was initiated during the master's thesis of Alejandro Hernández Cano at EPFL. It was supported as part of the Swiss AI Initiative by a grant from the Swiss National Supercomputing Centre (CSCS) under project ID a06 on Alps.

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

## A  Hyperparameters

We detail the selection of hyperparameters used in Table 6. For the case of FOG-flash, the $\alpha_0$ initialization value of $\tanh_\alpha$ entropy-regularization is 0.5 for all model sizes. All models use a linear warmup schedule, and 1-sqrt cooldown schedule. The long-data 1.5B FOG-max experiment was trained for a total of 430,000 steps, consuming approximately 450.9B tokens, using the same hyperparameters as the shorter run, including warmup steps.

| Hyperparameter | 390M | 1.5B | 8B |
|---|---|---|---|
| Layers ($L$) | 16 | 16 | 32 |
| Hidden size ($D$) | 1024 | 2048 | 4096 |
| FFN hidden size | 4096 | 8192 | 14336 |
| Attention heads | 8 | 16 | 32 |
| QK groups | 4 | 8 | 8 |
| Softmax scale* ($s$) | 0.17678 | 0.125 | |
| Tied embeddings | Yes | No | |
| Weight decay ($\lambda$) | | 0.1 | |
| AdamW $\beta_1$ | | 0.9 | |
| AdamW $\beta_2$ | | 0.95 | |
| Gradient clip value | | 1.0 | |
| Context length $T$ | | 4096 | |
| Global batch size | 128 | 256 | 512 |
| Total training steps | 100,000 | 125,000 | 10,000 |
| Peak learning rate ($\eta$) | $10^{-3}$ | $2.5 \times 10^{-4}$ | $1.5 \times 10^{-4}$ |
| Warmup $\eta$ steps | 5,000 | 2,500 | 1,250 |
| Cooldown $\eta$ steps | 20,000 | 25,000 | N/A |
| Minimum $\eta$ | | $10^{-8}$ | |

Table 6: **Hyperparameters used in experiments.** Note that FFN hidden size indicates the dimensionality of each linear projection in gated activation functions; networks without GLUs use $1.5\times$ this value to match the parameter count. Softmax scale specified only applies to FOG models, all other models follow the standard $s = 1/\sqrt{D_{QK}}$.

## B  Architectures

We provide detailed formulations for all architectures presented in this paper. Our transformer architecture consists of the following components in sequence:

1. Input token embeddings
2. An input scaling factor $u \in (0, \infty)$, which may equal 1
3. A series of $L$ transformer blocks as described below
4. A final normalization function $N^{(\text{final})}$, which may be the identity
5. A linear output layer

The transformer block is defined as

$$\text{block}(\mathbf{X}) := \hat{\mathbf{X}} + \left( N_2^{(\text{post})} \circ \text{FFN} \circ N_2^{(\text{pre})} \right)(\hat{\mathbf{X}}), \qquad \hat{\mathbf{X}} := \mathbf{X} + \left( N_1^{(\text{post})} \circ \text{GQA} \circ N_1^{(\text{pre})} \right)(\mathbf{X}).$$

The $N_i^{(*)}$ are normalization layers that may be the identity, and $\text{FFN}(\mathbf{X})$ is a two-layer FFN with a nonlinear activation function $\varphi$ and no bias. The GQA follows the standard grouped-query self-attention definition with softmax scaling factor $s$ and Rotary Position Embeddings. Each attention head uses the definition

$$\text{attnhead}(\mathbf{X}) := \text{selfattn}\left( N_Q^{(\text{QK})}(\mathbf{X}\mathbf{W}^{(Q)}), N_K^{(\text{QK})}(\mathbf{X}\mathbf{W}^{(K)}), \mathbf{X}\mathbf{W}^{(V)} \right),$$

where $N^{(\mathrm{QK})}$ is the entropy-regularization mechanism, and $\mathrm{selfattn} = \mathbf{PV}$. The $\mathbf{P}$ matrix is the attention probabilities matrix

$$\mathbf{P} := \mathrm{Softmax}\left(s\mathbf{QK}^\top + \mathbf{M}\right) \tag{1}$$

With this notation, Table 7 details the architecture families used in the project.

| Parameter | $u$ | $N^{(\mathrm{final})}$ | $N^{(\mathrm{pre})}$ | $N^{(\mathrm{post})}$ | $N^{(\mathrm{QK})}$ | $\varphi$ |
|---|---|---|---|---|---|---|
| Llama3 | 1 | $\mathcal{N}_\gamma$ | $\mathcal{N}_\gamma$ | id | id | SwiGLU |
| Llama3+SmoothSwiGLU | 1 | $\mathcal{N}_\gamma$ | $\mathcal{N}_\gamma$ | id | id | SmoothSwiGLU |
| OLMo2 | 1 | $\mathcal{N}_\gamma$ | id | $\mathcal{N}_\gamma$ | $\mathcal{N}_\gamma$ | SwiGLU |
| OP[a] | $\sigma_0^{-1}$ | id | id | $\mathrm{LayerScale}_\gamma$ | $\mathcal{N}_\gamma$ | GeLU |
| FOG-max[a,b] | $\sigma_0^{-1}$ | id | id | $\mathcal{N}_\gamma$ | $\mathcal{N}$ | xIELU |
| FOG-opt[a,b] | $\sigma_0^{-1}$ | id | id | $\mathcal{N}_\gamma$ | $\mathcal{N}$ | GeLU |
| FOG-flash[a,b] | $\sigma_0^{-1}$ | id | id | $\mathcal{N}_\gamma$ | $\tanh_\alpha$ | GeLU |

Table 7: **Architecture details for the used models.** Models with (a) initialize the post-normalization gains with $\gamma_0 = 1/\sqrt{L}$. Models with (b) have frozen gains in the QK entropy regularization $N^{(\mathrm{QK})}$. The $\mathrm{id}$ is the identity function, $\sigma_0$ is the chosen initialization standard deviation, $\mathcal{N}$ is the RMS normalization. The $u$ input scaling is not trainable.

**xIELU activation function**   Introduced in [13], the xIELU activation function is defined element-wise as:

$$\mathrm{xIELU}(x) := \begin{cases} \alpha_p x^2 + 0.5x & \text{if } x > 0, \\ \alpha_n(e^x - 1) - \alpha_n x + 0.5x & \text{if } x \leq 0. \end{cases} \tag{2}$$

where $\alpha_p$ and $\alpha_n$ are trainable scalars per layer. xIELU is an extension of Squared ReLU and has been adopted and validated at scale [11].

## C   Outliers impact quantization

Let's start with a useful definition.
$\tau$-**outlier**: Given $\mathbf{x} \in \mathbb{R}^d$ and $\sigma = \mathrm{rms}(\mathbf{x})$, the element $x$ of $\mathbf{x}$ is a $\tau$-outlier if $|x| \geq \tau\sigma$.

As $\tau$ increases, $x$ becomes a larger outlier ($\sigma$ represents the natural magnitude of $\mathbf{x}$). In practice, $\tau \gg 1$. Before FP8 quantization, each tensor is scaled with $s(\mathbf{x}) := \mathrm{MaxFP8Value}/\mathrm{absmax}(\mathbf{x})$ to better utilize limited FP8 dynamic range.

> **Theorem 1.** Let $\mathbf{x} \in \mathbb{R}^d$ have $\tau$-outlier $x_j$, and $\mathbf{x}' \in \mathbb{R}^d$ have a $\tau'$-outlier $x'_j$ with $\tau' > \tau$. Then for any subset $T \subseteq \{1 \ldots d\} \setminus j$, vector $\mathbf{x}'_T$ will be quantized less accurately than $\mathbf{x}_T$.

In other words, larger outlier values lead to less precise FP8-quantized results.

### C.1   Proof

Let $r = \mathrm{FP8MaxValue}$, $s = r/\mathrm{absmax}(\mathbf{x})$, and $s' = s(\mathbf{x}')$.

Since $\mathrm{absmax}(\mathbf{x}) \geq |x_j| \geq \tau\sigma$, then $s \leq \frac{r}{\tau\sigma}$ (similarly, $s' \leq \frac{r}{\tau'\sigma}$). Let $m = \mathrm{absmax}(\mathbf{x}_T)$ and $m' = \mathrm{absmax}(\mathbf{x}'_T)$.

Elements of $\mathbf{x}_T$ lie in $[-m, m]$. After scaling, the range becomes $[-rm/(\tau\sigma), rm/(\tau\sigma)]$ in $s\mathbf{x}_T$, and $[-rm/(\tau'\sigma), rm/(\tau'\sigma)]$ in $s'\mathbf{x}'_T$. Since $\tau' > \tau$, $[-rm/(\tau'\sigma), rm/(\tau'\sigma)] \subset [-rm/(\tau\sigma), rm/(\tau\sigma)]$, so $\mathbf{x}'_T$ has smaller range.
This narrowed range contains fewer n-bit representable numbers, proving the theorem.

The proof applies to any subset $T$, including the set of "typical" values (e.g., 90%-quantile).
Theorem 1 guarantees large outliers worsen quantization on 90% of tensor elements.

## C.2 Empirical confirmation

We measured activation values of Llama and FOG-max 1.5B during mid-training on a micro batch of data (precisely the second FFN layer's input, before quantization).

**Observation:** Llama presents a 688-outlier while FOG-max shows only a 183-outlier. Using 90%-quantile, we get that **90% of Llama's activation coefficients scale to** $[-0.289, 0.289]$ range, while **FOG-max allows a much broader range of** $[-2.084, 2.984]$.

# D FOG extensions

**FOG-SwiGLU** In addition to our main experiments, we trained a 1.5B FOG model using the SwiGLU activation function, which we label FOG-SwiGLU. This architecture was adapted from FOG-max, changing the activation function to SwiGLU and adjusting FFN hidden size to match the parameter count. Figure 8 shows the loss progression of this model under FP8DPA training. This experiment resulted in a completely stable training, and further demonstrates the flexibility of activation functions suitable in our design.

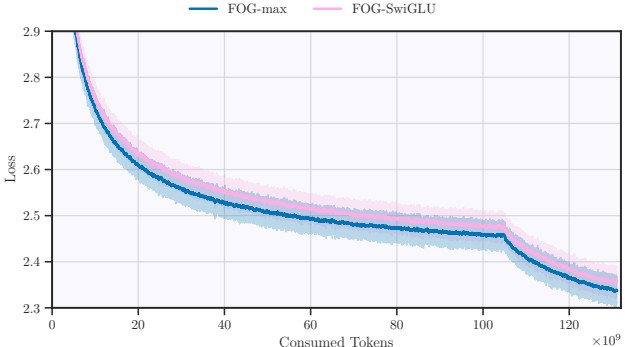

Figure 8: **FOG-SwiGLU 1.5B FP8DPA run.** FOG-max included as reference. We observe stable training dynamics for both approaches. The reported loss is the 200-rolling-window mean and 5%-95% quantile bands.

**MoE extension** We adapted FOG-flash architecture to follow an MoE design, keeping the backbone configuration of the 390M model (hidden size, number of layers, etc), but upscaling with 8 FFN experts (2 active) following [17], resulting in a 1.8B model, trained from scratch under the same configuration as all 390M models. Additionally, we employ the z-loss [17] with coefficient of 0.01, and no explicit loss-balancing loss, but rather an expert bias [33] with update rate of 0.01. We trained this model under BF16 and FP8DPA training, resulting on a final converged loss of 2.477 and 2.483, respectively, as shown in Figure 9. This stable training result suggests good generalization capabilities for FOG to other MoE designs under FP8DPA training. We further adapted our 1.5B models and similarly scale up to 8 experts (2 active) to measure throughput gains at a larger scale. Table 8 summarizes our results, where FOG still provides the most throughput gains.

| Model | Precision | Throughput (tokens/second/GPU) |
|---|---|---|
| Llama3 | BF16 | 3336 |
| Llama+SmoothSwiGLU | FP8 | 4202 (+24.8%) |
| FOG-flash | FP8DPA | **4351 (+29.2%)** |

Table 8: **MoE training throughput.** Measurements of 41B-8E MoEs taken using 4xGH200 nodes with expert parallel and pipeline parallel size of 4 using a batch size of 512. As with dense models, FOG-flash outperform all other architectures.

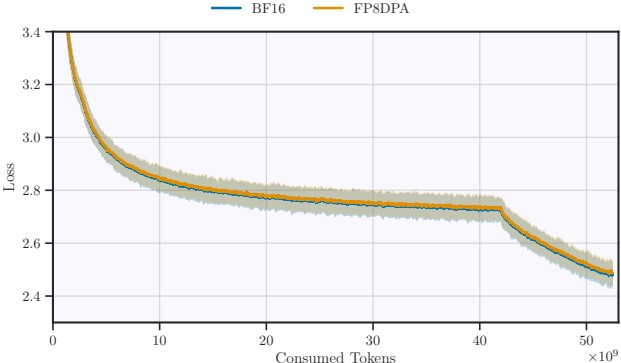

Figure 9: **FOG-flash-MoE 1.8B-8E FP8DPA loss progress.** Both BF16 and FP8DPA trainings are shown. The FP8DPA training remains stable for the entire duration of training, and the final loss converged to in both precisions remains within $\pm 0.005$, suggesting comparable downstream capabilities. Loss reported is the 200-rolling-window mean and 5%-95% quantile bands.

# E    FP8 training

In all our experiments, we used Transformer Engine's delayed scaling implementation with history length $\ell = 1024$ and margin $m = 0$. Mathematically, given a history of abs-max values, denoted $H = \{h_t\}_{t=1}^{\ell} \subseteq [0, \infty)$, of a tensor $\mathbf{X}$, we define its scaling factor as:

$$\rho(\mathbf{X}) := \frac{\text{FP8MaxValue}}{2^m \max H}$$

where $\text{FP8MaxValue} \in (0, \infty)$ is the maximum value representable with the FP8 format used. We update the history using $H \leftarrow \{\max_{x \in \mathbf{X}} |x|\} \cup \{h_t\}_{t=2}^{\ell}$ to use for this activation in the next iteration. The end-to-end FP8 matrix multiplication is

$$\text{GEMM}(\mathbf{X}, \mathbf{Y}) := \frac{1}{\rho(\mathbf{X})\rho(\mathbf{Y})}\text{FP8GEMM}(\text{FP8cast}(\rho(\mathbf{X})\mathbf{X}), \text{FP8cast}(\rho(\mathbf{Y})\mathbf{Y})),$$

where FP8GEMM receives FP8 tensors and returns the BF16 result. We further detail the precision used for every matrix multiplication during our FP8 and FP8DPA experiments in Table 9.

| Method | Linear operators | Attention scores $\mathbf{QK}^{\top}$ | Attention-value GEMM $\mathbf{PV}$ | Output layer |
|---|---|---|---|---|
| FP8 | FP8 | BF16 | BF16 | BF16 |
| FP8DPA | FP8 | FP8 | FP8 | BF16 |

Table 9: **Comparison between FP8 methods.** The FP8DPA method allows for all GEMM computations—excluding the output head— to be done with FP8 precision. In contrast, FP8 training uses higher precision for the core attention computation. The linear operators are linear layers of the form $\text{Linear}_{\mathbf{W}}(\mathbf{X}) = \mathbf{XW}$: namely the FFN linear layers, QKV projections and attention output projection. See Equation (1) for the definition of the attention probability matrix $\mathbf{P}$.

**FP8 optimizer moments**    To further reduce memory usage, we tested FOG-flash under the usual FP8DPA setting, with an additional constraint: FP8 optimizer moments. This extends the typical setting of half-precision gradients and moments used in most of our experiments. Figure 10 shows the training loss across three different settings for comparison.

**Fine-grained scaling recipes**    Recent FP8 training achievements, such as DeepSeek's DeemGEMM kernels, involve the use of fine-grained FP8 scaling recipes to provide a more robust training regime. While these options could potentially enable FP8 training when tensor-wise scaling alternatives diverge, it comes with a significant overhead. We validate this claim using TransformerEngine's

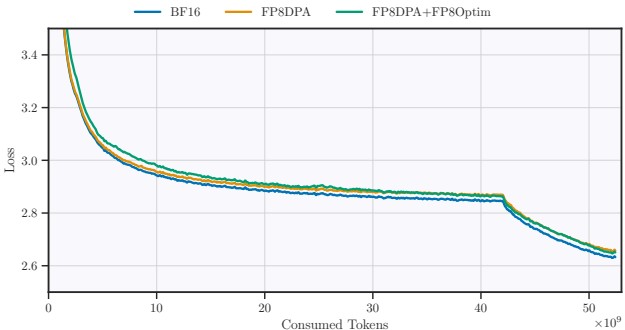

Figure 10: **FOG-flash 390M loss curve comparing training precision.** Our design reaches similar loss when trained with either precision. Loss reported is the 200-rolling-window mean.

Blockwise scaling at the 8B scale. Training throughput is reported in Table 10. Using FP8DPA training with delayed scaling recipe provides the highest boost across all tested methods.

| Model | Precision | FP8 Recipe | Throughput (tokens/second/GPU) |
|---|---|---|---|
| Llama3 | BF16 | N/A | 9.48k |
| Llama3 | FP8 | Blockwise | 11.18k (+17.9%) |
| OP | FP8 | Delayed | 12.14k (+28.1%) |
| Llama+SmoothSwiGLU | FP8 | Delayed | 13.1k (+38.2%) |
| FOG-flash | FP8DPA | Delayed | **13.52k (+42.6%)** |

Table 10: **Training throughput.** Measurements taken using eight 4xGH200 nodes with Zero-1 sharding [26], without model parallelism using a batch size of 1024. Notably FOG-flash outperform all other architectures.

# F   Long context

Note that when global batch size (GBS) increases –micro batch size fixed–, computation time takes over communication time. Therefore, the larger the GBS, the higher the throughput gains for all approaches: FOG-flash reaches +42.6% with GBS=1k compared to +40.2% with GBS=512. Moreover, enabling FP8 computations in the attention bring unique throughput benefits under long-context training. Hence the large efficiency gap achieved by FOG-flash FP8DPA compared to Llama3+SmoothSwiGLU in Table 11.

| Context Length | TP | Llama3 (BF16) | FOG-flash (FP8DPA) | Llama3+SmoothSwiGLU (FP8) |
|---|---|---|---|---|
| 4096 | 1 | 9.48K | 13.52K | 13.13K |
| 8192 | 1 | 9.08K | 13.03K | Out Of Memory |
| 8192 | 2 | 7.49K | 10.42K | 10.05K |
| 16384 | 2 | 6.81K | 9.45K | 8.93K |

Table 11: **Training throughput under varying sequence lengths**, performed at 8B scale using eight 4xGH200 nodes with a global batch size of 1024. TP refers to Tensor Parallelism.

# G   Evaluations

We selected the following set of benchmarks: `ARC-Easy`, `CommonsenseQA`, `HellaSwag`, `LAMBADA-OpenAI`, `LAMBADA-standard`, `OpenBookQA`, `PIQA`, `SocialIQA`, and `WinoGrande`. We

used a standard open-source LLM evaluation package for conducting these evaluations, as cited in the code repository `https://github.com/anonymous4375934/FOG`.

In Table 5, we report raw accuracy scores as percentages on three key benchmarks as well as the average over the full set of tasks mentioned above. In Table 12, we provide all scores along with their estimation errors for the 1.5B model size, demonstrating that the slight differences observed across many values are statistically insignificant.

| Architecture | Llama3 | FOG-max | FOG-opt | FOG-flash | |
|---|---|---|---|---|---|
| Hellaswag | 43.7 \| − | 43.3 \| 43.4 | 43.3 \| 42.7 | 42.8 \| 41.9 | ±0.5 |
| ARC-easy | 71.8 \| − | 71.6 \| 73.0 | 71.3 \| 70.8 | 70.9 \| 69.4 | ±0.9 |
| PIQA | 72.5 \| − | 72.6 \| 73.3 | 72.5 \| 72.0 | 72.2 \| 72.0 | ±1.0 |
| Commonsense-qa | 19.6 \| − | 20.2 \| 22.2 | 19.3 \| 21.2 | 21.1 \| 20.8 | ±1.2 |
| Lambada-openai | 44.5 \| − | 43.7 \| 44.6 | 44.3 \| 44.5 | 42.4 \| 41.2 | ±0.7 |
| Lambada-standard | 38.9 \| − | 37.0 \| 39.5 | 37.7 \| 37.9 | 35.9 \| 33.8 | ±0.7 |
| Openbook-qa | 26.2 \| − | 28.0 \| 27.0 | 26.8 \| 27.0 | 28.4 \| 27.6 | ±2.0 |
| Social-iqa | 41.3 \| − | 41.8 \| 42.0 | 41.7 \| 40.8 | 41.0 \| 40.8 | ±1.1 |
| Winogrande | 56.5 \| − | 55.6 \| 58.7 | 54.6 \| 57.4 | 56.9 \| 56.8 | ±1.4 |
| Average | 46.1 \| − | 46.0 \| **47.1** | 45.7 \| 46.0 | 45.7 \| 44.9 | ±0.3 |

Table 12: **More detailed results at 1.5B scale.** For each model and each task, the first score results from BF16 training and the second from FP8DPA training.

**Weight decay cooldown** As mentioned in Section 3, we experimented with cooling down the weight decay, often used as a constant value equal to $0.1$ that is coupled with the learning rate, to see if it solves the OP+frozenQK architecture's consistent divergence during the learning rate decay phase. We also tested it on other architectures and, to optimize the use of resources, we had to keep it later for the final experiments. This trick helped stabilize the weights' norm indeed, but couldn't solve the divergence issue. Moreover, it had no effect on final performance nor on stability. Table 13 highlights this no-effect claim at 1.5B scale.

| Setting | WD | Loss | Average score |
|---|---|---|---|
| OP+FrozenQK | cooldown | diverges | - |
| OP+FrozenQK | constant | diverges | - |
| FOG-opt | cooldown | converges | 46.0 ±0.3 |
| FOG-opt | constant | converges | 46.3 ±0.3 |

Table 13: **Weight Decay (WD) during the LR decay phase**. If constant, it equals $0.1$. Else, it starts from $0.1$ and is proportional to LR.

# H    Computational Resources

Our experiments were conducted on nodes equipped with **4** Grace Hopper (GH200) GPUs each. We typically used 4, 8, and 16 nodes for our 390M, 1.5B, and 8B parameter experiments, respectively, with minor variations across different runs. Importantly, all throughput measurements were taken under identical hardware configurations. Table 14 details the computational resources in GPU hours (GPUh) required for our main experimental results. This includes the computational cost of training all architectures that diverged during FP8DPA training, the FP8DPA and BF16 stable training runs for our three main architectures, and the BF16 Llama3 baseline. The aggregation includes node start-up times, computation lost due to node failures, and overhead from calculating and logging kurtosis metrics. The complete research project required additional computational resources beyond those specified in the table, as we conducted numerous preliminary experiments and explored ideas that did not appear in the final paper.

| Group | GPUh |
|---|---|
| Divergent runs (FP8DPA) | 886 |
| Llama3 baselines (BF16) | 1,395 |
| FOG experiments | 11,162 |

Table 14: **GPU hours used for the main experiments.**

