# OpenReview forum: "Towards Fully FP8 GEMM LLM Training at Scale"
_NeurIPS.cc/2025/Conference — NeurIPS 2025 poster_

### Official Review · Reviewer_5MCk · 2025-06-29

**Clarity:** 3
**Significance:** 3
**Originality:** 2
**Rating:** 3
**Confidence:** 3

**Summary:**

This paper mainly addresses the stability issue of FP8 during the LLM pre training phase. The paper implemented FP8 training on Self attention and FFNs, and compared it with previous methods. Extensive experiments have demonstrated the effectiveness of the method. The core goal is to eliminate outlier issues that occur during the FP8 stage.

**Questions:**

See weaknesses.

**Ethical Concerns:**

["NO or VERY MINOR ethics concerns only"]

**Final Justification:**

I have re-examined this paper and will stand by my previous comments.

**Quality:**

3

**Strengths And Weaknesses:**

**Strengths**

1. This paper is well written, logically clear, and fully expresses the motivation and methods of the paper.
2. The paper conducts extensive experiments, including comparative experiments on models of different types and scales.
3. The authors provide a wide range of comprehensive evaluation metrics.

**Weaknesses**

1. Firstly, the study lacks necessary ablation experiments, such as the different components listed in Table 1. The author only provided conclusions without relevant ablation experiments. At the same time, there is a lack of more possibilities, such as activations outside of GeLU.

2. Secondly, for the modification of transformers, there are more enlightening and lack additional explanations for replacing or deleting different components.

3. Finally, the core purpose of the article is to eliminate the negative impact of outliers. The author only presented the final results and lacked a key explanation for eliminating outliers. If there could be a more reasonable explanation, it would enhance the completeness of this article.

---

> ### Author Rebuttal · Authors · 2025-07-31
>
> We thank the reviewer for their feedback and hope the concerns will be fully addressed with the following response.
>
> > The study lacks necessary ablation experiments | The modification of transformers
>
> We would like to point the reviewer to the Section 3 (after Table 1) as well as Section 4, where we present **extensive ablations and a step-by-step transition** from OP architecture, that faces early divergence, to FOG-Max variant. We further **validate with kurtosis observations** (Figure 3) to isolate the components causing instability or outlier amplification. Including Section 5, we have **ablated around 8 architectures, covered 3 model sizes (0.4B, 1.5B, 8B) and different data regimes (50BTs, 125BTs, 420BTs)**. We would also like to quote the reviewer 5MCk: "The paper conducts extensive experiments, including comparative experiments on models of different types and scales" and "The authors provide a wide range of comprehensive evaluation metrics".
>
> To further address the reviewer's concern, **we conducted additional experiments, this time starting from Llama architecture and reaching a new variant we call FOG-SwiGLU**. In this study, QK norm is introduced after Post-Layernorm, considerably delaying divergence. The delayed divergence confirms the benefits from its entropy regularization effect, avoiding peaky softmax outputs. Finally, Freezing QK ensures convergence while matching the Llama3 BF16 baseline.
> The following table presents further details about the study:
>
> | Size=390M (FP8DPA)     | Data Budget | Normalization | QK-RMSNorm | Activation Function | Observation                      |
> |-------------------------|-------------|----------------|---------|----------------------|----------------------------------|
> | Llama3                  | 50B tokens  | Pre-LN         | No      | SwiGLU               |    Early divergence (Fig-6)        |
> | Llama3 w/ Post-LN        | 50B tokens  | Post-LN       | No      | SwiGLU               |   Early divergence (new) : lower kurtosis |
> | oLMo2                     | 50B tokens  | Post-LN       | Yes     | SwiGLU               | Mid-training divergence (Fig-6)  |
> | FOG-SwiGLU-1.5B**        | 125B tokens | Post-LN       | Yes *   | SwiGLU               | Convergence, equivalent to baseline (new) |
>
> *Frozen QK-norm gains
> **Note that the FOG-SwiGLU ablation scales up model size and data budget compared to the other 390M architectures.
>
> Reviewer also mentioned that “ there is a lack of more possibilities, such as activations outside of GeLU “.
> We remind that FOG-Max, a key variant among the initially proposed FOG architectures, uses **xIELU activation function** which belongs to the family of quadratic and learnable activation functions. The above-mentioned additional study, further validates the stability of a 4th variant that simply keeps **SwiGLU gated activation function**. This demonstrates unprecendented generalization, unlike prior work which is tied to a scaled version of gated activations
>
> > Lacked a key explanation for eliminating outliers
>
> The core motivation of eliminating outliers stems from the fact that, because of the limited dynamic range of FP8 formats, the presence of large outlier features increases the risk of under- or over-flowing values.
> This is motivated in lines 26-32, 89-97, 158-170 as well as the references cited therein [1,2,3,4].
> Moreover, our experiments support that mitigating large outliers is critical to enable FP8 training.
> - In Figure 3, large activation outliers were present on during cooldown, resulting on divergence of an otherwise stable FP8DPA training run.
> - Figure 5 demonstrates large activation outliers prior to the divergence of an FP8DPA run.
>
> Speaking about outliers, this brings the topic of outlier amplification profoundly studied on Llama by the prior work [4], introducing the smooth SwiGLU variant. It shows FP8 outliers are amplified by SwiGLU that can act quadratically late in training (cf. Lines 183-191).
> One of our work's novelties is allowing FP8**DPA** training with **not only** scaled gated activations but a wide range of non-linearities stemming sub-linear in behaviour (GeLU),  Gated (SwiGLU), and especially quadratic activation functions (xIELU).
> The fact that such a quadratic activation function works smoothly with FOG architecture, as evidenced in the Table addressing the previous point in this rebuttal, although it is expected to amplify outliers, is **an important signal in favour of FOG's robustness**.
> We aim to make this point, currently raised in Line 183, clearer in the next revision.
>
> [1]: Tim Dettmers, Mike Lewis, Younes Belkada, Luke Zettlemoyer, 2022, LLM.int8(): 8-bit Matrix Multiplication for Transformers at Scale, in Advances in Neural Information Processing
> Systems.
> [2]: Mingjie Sun, Xinlei Chen, J Zico Kolter, Zhuang Liu, 2024, Massive Activations in Large Language Models, in First Conference on Language Modeling.
> [3]: Bobby He, Lorenzo Noci, Daniele Paliotta, Imanol Schlag, Thomas Hofmann, 2024, Understanding and Minimising Outlier Features in Neural Network Training, in The Thirty-eighth Annual Conference on Neural Information Processing Systems.
> [4]: Maxim Fishman, Brian Chmiel, Ron Banner, Daniel Soudry, 2025, Scaling FP8 training to trillion-token LLMs, in The Thirteenth International Conference on Learning Representations.

---

> > ### Comment · Reviewer_5MCk · 2025-08-06
> >
> > Thank you for your reply, which has deepened my understanding of the paper. Regarding the elimination of outliers, I have not seen more positive evidence or theoretical support; it tends to be heuristic. I prefer to maintain the original score.

---

> > > ### Author Response · Authors · 2025-08-06
> > >
> > > We thank the reviewer for their feedback. However, we respectfully disagree with the rejection recommendation for multiple reasons.
> > >
> > > Due to the character limit, we have to split our response into two main parts.
> > > Initially, we will defend the already provided motivation and give more feedback.
> > > In the next comment, we will provide two additional theoretical arguments with new empirical validation, further detailing and strengthening the outlier minimization motivation.
> > >
> > > ## Motivation
> > > First, regarding outlier minimization:
> > > - **Our inductive approach**: The novel FP8DPA training achievement, proven linked to outlier minimization through kurtosis analysis, provides empirical evidence of this procedure's importance. We've also shown standard architectures like Llama (Figure 4) and OLMo (Figure 5) face divergence due to higher kurtosis levels (i.e more or/and larger outliers).
> > > - **Our deductive approach**: As mentioned in the rebuttal, we motivated outlier minimization under low precision training by FP8's limited dynamic range, that leads to more frequent underflows, overflows, and generally severe quantizations errors.
> > > - **References** (respectively [3] and [2] in the manuscript) provide similar motivations:
> > >     -  **[1]** : *"The main challenge with quantization methods using a single scaling constant per tensor is that a single outlier can reduce the quantization precision of all other values"*.
> > >     - DeepSeek-v3 **[2]** confirms: *"In low-precision training frameworks, overflows and underflows are common challenges due to the limited dynamic range of the FP8 format, which is constrained by its reduced exponent bits"*.
> > >
> > > ## Feedback
> > > Second, more generally:
> > > - **Successful rebuttal**: All weaknesses except remaining motivation skepticism were fully addressed. We've proven incorrect the comments about lacking: *"necessary ablation experiments," "activations outside of GeLU", "additional explanations for replacing or deleting different components"*. In addition, the reviewer finds the paper "well written, logically clear, and fully expresses the motivation and methods."
> > > - **Objective feedback**: Both reviewers who deeply engaged in discussions have improved evaluations, leaning toward clear accepts, trusting this work "will benefit many researchers" (reviewer r2ET). Discussions revealed several **new strengths**, which we kindly invite the reviewer to consider for a fuller picture:
> > >     - Generalization to MoE designs.
> > >     - Even greater benefits in long context scenarios (attention-bottelnecked)
> > >     - Stability and maintained performance with FP8 optimizer states.
> > >
> > > [1] Dettmers et al., "LLM.int8(): 8-bit Matrix Multiplication for Transformers at Scale," arXiv:2208.07339, 2022.
> > >
> > > [2] DeepSeek‑AI et al., "DeepSeek‑V3 Technical Report," arXiv:2412.19437, 2024.

---

> ### Author Response · Authors · 2025-08-07
>
> ## Additional arguments and experiments
> We understand the reviewer seeks more detailed **deductive** analysis of outlier minimization. Therefore, **we provide two additional arguments with theoretical and empirical analysis** of negative consequences when such features are present.
>
> **Argument 1: Outliers fundamentally lead to less-precise quantization**
>
> Let's start with a useful definition. $\tau$-outlier: Given $\mathbf{x} \in \mathbb{R}^d$ and $\sigma = \mathrm{rms}(\mathbf{x})$, element $x$ of $\mathbf{x}$ is a $\tau$-outlier if $|x| \geq \tau \sigma$.
>
> As $\tau$ increases, $x$ becomes a larger outlier ($\sigma$ represents the natural magnitude of $\mathbf{x}$).
> In practice, $\tau \gg 1$.
>
> Before FP8 quantization, each tensor is scaled with $s(\mathbf{x}) := \mathrm{MaxFP8Value}/\mathrm{absmax}(\mathbf{x})$ to better utilize limited FP8 dynamic range.
>
> **Theorem 1**
> > Let $\mathbf{x} \in \mathbb{R}^d$ have $\tau$-outlier $x_j$, and $\mathbf{x}'$ identical to $\mathbf{x}$ except $x_j'$ is $\tau'$-outlier with $\tau' > \tau$.
> >
> > Then for any subset $T \subseteq \\{1\ldots d\\} \setminus \{j\}$, vector $\mathbf{x}_T'$ will be quantized less accurately than $\mathbf{x}_T$.
>
> In other words, larger outlier values lead to less precise FP8-quantized results.
>
> **Proof**
> > Let $r = \mathrm{FP8MaxValue}$, $s = r/\mathrm{absmax}(\mathbf{x})$, and $s' = s(\mathbf{x}')$.
> Since $\mathrm{absmax}(\mathbf{x}) \geq |x_j| \geq \tau \sigma$, then $s \leq \frac{r}{\tau \sigma}$ (similarly, $s' \leq \frac{r}{\tau' \sigma}$).
> Let $m = \mathrm{absmax}(\mathbf{x}_T)$ and $m' = \mathrm{absmax}(\mathbf{x}_T')$.
> Elements of $\mathbf{x}_T$ lie in $[-m, m]$.
> After scaling, range becomes $[-rm/(\tau \sigma), rm/(\tau \sigma)]$ in $s \mathbf{x}_T$, and $[-rm/(\tau' \sigma), rm/(\tau' \sigma)]$ in $s' \mathbf{x}_T'$.
> Since $\tau' > \tau$, $[-rm/(\tau' \sigma), rm/(\tau' \sigma)] \subset [-rm/(\tau \sigma), rm/(\tau \sigma)]$, so $\mathbf{x}_T'$ has smaller range.
> This narrowed range contains fewer n-bit representable numbers, proving the theorem.
>
> The proof applies to any subset $T$, including the set of "typical" values (e.g., 90%-quantile). Theorem 1 guarantees large outliers worsen quantization on 90% of tensor elements.
>
> **Empirical confirmation**
> We measured activation values of Llama and FOG-max 1.5B during mid-training on a micro batch of data, (precisely the second FFN layer's input, before quantization). `Observation`: Llama presents a 688-outlier while FOG-max shows only a 183-outlier. Using 90%-quantile, we get that **90% of Llama's activation coefficients scale to** $[-0.289, 0.289]$ range, while **FOG-max allows a much broader range of** $[-2.984, 2.984]$.
>
> **Argument 2: Outlier Cross-Term and Saturation**
> > Standard architectures contain outliers in weights and activations carrying critical information [1, 2]. During FP8 GEMMs, the informative products outlier x outlier terms typically exceed FP8's range, causing information loss.
>
> `Proof`:
> This stems from FP8's limited dynamic range (E4M3: max normal magnitude ±448). Theoretically, multiplications like 447x440; 5x200; 20x25 would yield infinity. Even if the accumulator is of higher precision, such products will likely cause clamping to the same maximum value=448 (saturation) later during casting, erasing magnitude differences.
> In per-tensor scaling, outliers cause the scaling factor to be small ($s=448/amax$). As un‑scaling in subsequent operations
> scaling uses the inverse (large) factor, this leads to more outliers emerging downstream: multiplication by $1/s$ then inflates all saturated entries to match the magnitude of the original outliers. This propagation of large values can create new outliers downstream and, combined with the loss of magnitude information, can disrupt training dynamics and in severe cases cause divergence.
>
> `Empirically`, the FFN final output from previous argument shows: **90% of FOG-max elements scale to** $[-11.75, 11.75]$—much larger and **better FP8-representable than Llama's** $[-0.617, 0.617]$ range.
>
>
> **Finally** we hope the reviewer is now convinced that this work's motivation is technically solid and kindly ask for re-evaluation given these theoretical and empirical explanations. We're happy to continue discussion if new concerns arise.
>
> [1] Bondarenko et al., "Outlier Dimensions that Disrupt Transformers are Driven by Frequency," EMNLP, 2021.
> [2] Dettmers et al., "LLM.int8(): 8-bit Matrix Multiplication for Transformers at Scale," arXiv:2208.07339, 2022.

---

### Official Review · Reviewer_nVZB · 2025-07-02

**Clarity:** 3
**Significance:** 2
**Originality:** 3
**Rating:** 4
**Confidence:** 4

**Summary:**

This work enables fully FP8 training of large language models (LLMs), including all GEMMs, even in attention layers (termed FP8DPA), to improve training throughput without sacrificing performance. It proposes the FOG architecture with changes in norm and activation layers. Besides, the kurtosis metric is proposed to track the outliers in the activations to diagnose/predict potential divergence behavior. This work leads to up to 40% speedup compared to BF16 baselines.

**Questions:**

There's a statement in L34 that multiple scaling factors per single tensor"comes with an efficiency overhead". Do you have benchmark result to demonstrate how much slower it is? I believe the deepseek deepgemm kernels are already heavily optimized with cutlass and low level primitives.

**Ethical Concerns:**

["NO or VERY MINOR ethics concerns only"]

**Final Justification:**

The additional experiments addressed most of my concerns:
FP8 optimizer states - addressed
Higher precision for the final projection layer - confirmed as limitation
Limited generality with regard to other architectures - partially addressed with preliminary results
Sequence length beyond 4k - addressed
Lack of clarity in Table 4 - addressed

**Limitations:**

- FOG’s techniques may not generalize to all future transformer variants. Specifically, it does not test the MOE model family, and it's unclear if it's activation change would affect the performance of MoE models.
limitations mentioned by the author:
- FP8 optimizer states not yet explored.
- it still uses higher precision for the final projection layer due to sensitivity.

**Quality:**

2

**Strengths And Weaknesses:**

Strength:
- the first work to enable fp8 training on attn (and hence all GEMMs)
- the kurtosis is a novel, insightful metric to predict outlier and potential divergence issues, with experimental result and analysis done on fog and non-fog model recipes.
- addresses an important problem since fp8 dtype is available in mainstream accelerators from AMD and NVIDIA
- reasonable speedup and convergence result

Weakness:
- the benefit of fp8 attention is not clarified. It does not show breakdown analysis on how much efficiency benefit fp8 attn brings, compared to other fp8 baselines such as OP. It would be good to justify the motivation to run fp8 attn.
- In this work, only 4k context window is used. Due to computation pattern of attn, the longer the sequence length, the more flops it consumes (hence more benefit if it's done in fp8). However, this work does not explore sequence length longer than 4k, which brings questions on whether the technique is useful for scenarios where bf16 attention computation is truly the bottleneck of the training. Furthermore it is fundamentally harder to run fp8 attention with longer sequences due to numerical stability issues, which I'd like to see it addressed.
- more rigorous benchmark is needed. The evaluation datasets (hellaswag/ARC/etc) are relatively simple. More comprehensive evaluation would make this work more convincing.
- Lack of clarify in table 4. Does Llama+SmoothSwiGLU mean bf16 attn? If not, there is missing reasonable fp8 baseline that converges. For instance, assuming deepseek-v3 fp8, and llama+SmoothSwiGLU+bf16 attn are expected to convergence, and FOG should be comparing with these converging fp8 baselines, on the efficiency gain. If Llama+SmoothSwiGLU does use bf16 attn, then it brings another question that fp8 attn does not bring much actual benefit (efficiency) compared to already working fp8 recipes.
- limited generality with regard to other architectures, such as MoE

---

> ### Author Rebuttal · Authors · 2025-07-31
>
> We thank the reviewer for the submitted review and weaknesses highlighted.
> Please find next a detailed answer for each point raised next.
>
> > The benefit of fp8 attention
>
> The main advantage of FP8DPA training in our work is the significant throughput boost compared to existing methods, as shown in Table 4.
> Moreover, we kindly refer the reviewer to the answers of the next points in the current rebuttal, where we include new experiments comparing other FP8 recipes, and efficiency on longer sequence lengths.
> We believe these additions further strengthen the advantages of FP8DPA training.
>
> > Sequence length beyond 4k
>
> We thank reviewer for this excellent point.
> First, we note that the sequence length of 4096 adopted in our experiments lies within the standard configurations of current small and large scale language model training setups such as DeepSeek-v3, OLMo2, Qwen2.5, and SmolLM3.
> This pre-training stage usually takes the largest amount of time, mainly because of the attention's quadratic complexity. For instance, only ~100B out of 11.3T of SmolLM3, and 0.8T out of the nearly 16T of Llama3.1 405B tokens were consumed in the long-context extension training phase.
> Hence, it matters the most to speed up this compute-consuming moderate-context phase of training.
>
> Nonetheless, long context training phase can still be relatively expensive.
> Therefore, by safely performing attention computation in FP8, our novel approach benefits the most from context extension.
> Indeed, attention tends to dominate the computational cost of the forward-backward pass as sequence length increases, which leads to even larger efficiency gains when using FP8DPA compared to previous FP8 approaches.
> **We demonstrate this trend at 8B model scale**  in the following throughput (tokens/second/gpu) table:
>
> | Sequence Length | LLaMA BF16 | FOG FP8DPA | Smooth-SwiGLU FP8 |
> | --- | --- | --- | --- |
> | 4K  (TP=1) | 9.48K | 13.52K (**+42.6%**) | 13.13K (**+38.5%**) |
> | 8K  (TP=1) | 9.08K | 13.03K (**+43.5%**) | Out Of Memory |
> | 8K  (TP=2) | 7.49k | 10.42K (**+39.1%**) | 10.05K (**+34.2%**) |
> | 16K (TP=2) | 6.81k | 9.45K (**+38.8%**) | 8.93K (**+31.1%**) |
>
> A batch size of 1k was used for all experiments, providing even better boosts than the ones reported in the paper (+40.2% --> +42.6%).
> Clearly, the gap between the FP8-only approach (Smooth-SwiGLU) and our FP8DPA setting becomes larger as context increases.
> We note that training with even longer contexts usually relies on strategies like ring attention, which bring back the sequence length processed per GPU to values closer to 16K. We thus limit ourselves to ablations of up to 16K.
>
> To address the concern about FOG's robustness in such long context scenarios, **we conducted two additional experiments** (FOG-Max FP8DPA; Llama BF16) at 390M scale for 50BTs, using **a sequence length of 16384**.
> Results can be summarized as follows: A similarly smooth loss progression with an equivalent final performance, around ~2.28 for FP8DPA FOG and 2.22 for BF16 Llama. Such a small loss gap may not translate to downstream performance, especially considering the possibility of running FOG long-context training for +38.8% more tokens at the same FLOP budget.
>
> > More rigorous benchmark is needed
>
> We understand the reviewer's concern about the "easy" benchmarks used in the evaluation.
> We first draw the attention of the reviewer to Table 4 of the appendix, where a wider set of benchmarks is detailed.
> While most of these are known to be "early signal" tasks, that is exactly the reason why we chose them:
> - The scale of 390M models trained on 50BTs and 1.5B models on 125BTs is **not large enough to expect for advanced capabilities to emerge**. Therefore, evaluating on "harder" tasks would result in close-to-random scores.
> - The selected tasks evaluate a **wide arrange of skills** (e.g. hellaswag for common-sense inference, OpenBookQA for assessing understanding of different subjects, etc).
> - A quick analysis of the scores shows that these **benchmarks are far from being saturated** by the trained models while exceeding the random performances. Thus, the comparison is fair for the scale.
> - Prior work [1] report performance on a similar suite of benchmarks, supporting their relevance in this context.
>
> > Lack of clarity in Table 4 | Granular FP8 recipes overhead
>
> The throughput reported in Table 4 for Llama+SmoothSwiGLU indeed utilizes BF16 computation in the attention, as enabling FP8 attention results in an early divergence shown in Figure 6.
> As explained in the Introduction and furthered detailed in Table 3 of the appendix, we associate the label "FP8" to the setting where attention is in BF16 and label "FP8DPA" the settings were FP8 attention computation is used.
> Next, **we include simple FP8 training with OP architecture and granular FP8 approaches** as these were not shown to diverge  and **remain viable FP8 candidates**.
> We compared the throughput of a set of 8B models with the same batch size (1k) across 8 nodes in the following table.
>
> | Architecture | Precision | FP8 recipe | Throughput (tokens/sec/gpu) |
> | :- | :-: | :-: | :-: |
> | Llama | BF16 | N/A | 9.48k |
> | Llama | FP8 | Blockwise* | 11.18k (+17.9%) |
> | OP | FP8 | DelayedScaling | 12.14k (+28.1%) |
> | Llama+SmoothSwiGLU | FP8 | DelayedScaling | 13.1k (+38.2%)  |
> | FOG-flash | FP8DPA | DelayedScaling | 13.52k (+42.6%) |
>
> \* NVIDIA's Transformer Engine implementation.
>
> Our approach still yields the highest efficiency.
> Responding to the concern that "fp8 attn does not bring much actual benefit (efficiency) compared to already working fp8 recipes", we argue, as motivated in the lines 33-35 of the paper and further evidenced by the table above, that using granular scaling factors such as  NVIDIA blockwise recipe, provides sub-optimal throughput gains. Such FP8 kernels are already heavily optimized but the overhead is inevitable from such a scaling strategy.
>
> Moreover, while the extra ~4.4% throughput boost compared to the fastest competing FP8 alternative (Llama+SmoothSwiGLU) could be seen as small, the proposed FOG architectures still have several advantages compared to it.
> Our FOG architecture family admits a **several QK regularization functions, and flexible choice of activation functions**, generalizing even to MoEs as demonstrated next.
> Moreover, the benefits of FP8DPA training become **more significant as the sequence length grows**, as shown previously, making our recipe more valuable during long-context training.
>
> In addition of GeLU and xIELU activations validated in the paper, **we additionally experimented with a 1.5B FOG-max FP8DPA variant with SwiGLU activation**, which we train for 125B tokens.
> This new experiment resulted in a **successful convergence**, similar to the BF16 baseline.
> Moreover, the choice of GeLU and xIELU activations in the proposed variants was deliberate and further validates that FOG can generalize to other activations. xiELU is a quadratic function and thus amplifies input outliers.
> Therefore, succesful FP8DPA training highlights FOG's robustness to outliers, and draws new limits of previous work [1], as discussed in lines 183-191.
> Moreover, the SmoothSwiGLU trick is not enough for stable FP8DPA training.
> We believe that allowing the use of quadratic activation functions with FP8DPA training is an important novelty of our work that we should make clearer in the next revision.
>
> > Limited generality with regard to other architectures
>
> The architecture suite proposed encompass various architectural choices with different normalization and activation function choices, as furthered evidenced in the previous point.
> While future transformer variants are impossible to predict fully, we provide state-of-the-art results for contemporary architectures and allow for flexibility within our architecture.
> Moreover, a key contribution of our work highlights the importance of kurtosis and outlier minimization for FP8 training monitoring, bringing a valuable tool to the scientific community to diagnose FP8 acceleration when such future transformer variants are proposed.
>
> While we don't provide extensive MoE results, we expect FOG to generalize well. Aside from the routing mechanism, MoEs follow similar forward passes as dense models. Our key changes, post-norm and QK regularization, don't interfere with MoE-specific components, and new FOG-SwiGLU stable variant further supports compatibility of our appraach.
> Still, **we adapted FOG-flash-390M to an MoE with 2 active experts among 8** (resulting in a 1.8B model). We trained it for 25BTs, following the experimental setup in our paper, with both FP8DPA and BF16 precision.
> The resulting 1.8B-8E FOG model converged to a loss of **2.477** with BF16, and **2.483** with FP8DPA. Which **suggests good generalization of FOG for MoEs under FP8DPA regime**.
>
> Finally, **we performed throughput measurements on a 41.5B-8E model**.
> As with dense models, our architecture provides the largest boost compared with other approaches.
>
> | Architecture | Precision | Throughput (tokens/sec/gpu) |
> | :- | :-: | :-: |
> | Llama MoE | BF16 | 3366 |
> | Llama+SmoothSwiGLU MoE | FP8 | 4202 (+24.8%)  |
> | FOG-flash | FP8DPA | 4351 (+29.2%) |
>
> > FP8 optimizer states
>
> In addition to the points discussed in the Limitations section, **we ran a new FP8DPA experiment** using FOG-flash at 390M scale **with FP8 optimizer moments**, following the setup in our paper.
> The model converged successfully to a loss of **2.645**, nearly identical to **2.649** obtained with higher-precision moments.
>
> > Higher precision for the final projection layer
>
> As mentioned in the Limitations section, we decided to explore this further in a future work.
> We believe the final projection layer is the most sensitive part of the architecture.
>
> [1]: Maxim Fishman, Brian Chmiel, Ron Banner, Daniel Soudry, 2025, Scaling FP8 training to trillion-token LLMs, in The Thirteenth International Conference on Learning Representations.

---

> > ### Author Response · Authors · 2025-08-05
> > **Clarification about the additional long context experiment**
> >
> > We wish to clarify an error in the additional long context experiment reported above, aimed at demonstrating stability and maintained performance. Upon review, we discovered that our BF16 baseline Llama model was mistakenly configured as **Llama-486M** (26% larger than intended) due to missing the FFN hidden dimension reduction when switching from pointwise to gated activation.
> >
> > After correcting the baseline to match the intended 390M parameters (FFN hidden dimension: 4096 instead of 6144), the **corrected results are**:
> > - FOG-max FP8DPA: **2.28**
> > - Llama BF16: **2.29** (correcting the erroneous 2.22)
> >
> > Therefore, FOG-max actually **slightly outperforms** the baseline, which aligns with the 4k context length results in the original manuscript and further supports FOG's robustness in long context scenarios. After checking, this was the only occurrence of this oversight among all original and additional experiments.

---

> > > ### Author Response · Authors · 2025-08-07
> > >
> > > Dear Reviewer nVZB,
> > >
> > > With nearly 25% of the discussion period remaining, we kindly invite you to share any feedback or comments you may have regarding our rebuttal. In it, we addressed your constructive critiques with detailed explanations and additional results that revealed several new strengths (MoE, Long context extra benefits, FP8 moments, etc). We also encourage you to consider the discussions with other reviewers for a fuller perspective.
> > > We respectfully request that you reconsider the evaluation in the light of these elements.
> > >
> > > Finally, we remain available for any further questions.
> > >
> > > Thank you for your time and effort.

---

> > > > ### Comment · Reviewer_nVZB · 2025-08-07
> > > >
> > > > Thanks the author for the detailed rebuttal. The additional experiments addressed my concern. I'll raise my score to 4 (borderline accept). Thank you for the efforts.

---

> > > > > ### Author Response · Authors · 2025-08-09
> > > > >
> > > > > Dear Reviewer,
> > > > > We appreciate the time you devoted to evaluating our manuscript, and thank you for the constructive feedback you provided. We are pleased that we were able to address all the concerns you raised, and we are grateful for your re-evaluation.

---

### Official Review · Reviewer_r2ET · 2025-07-03

**Clarity:** 2
**Significance:** 3
**Originality:** 3
**Rating:** 4
**Confidence:** 5

**Summary:**

This paper proposes an FP8 LLM training method.

**Questions:**

See weakness.

**Ethical Concerns:**

["NO or VERY MINOR ethics concerns only"]

**Final Justification:**

The authors have provided experimental evidence that sufficiently clarifies the loss behavior in extended-context scenarios. While I maintain that the scale of validation and longer context results require more extensive verification, I will recommend this paper for publication.

**Limitations:**

See Weakness.

**Paper Formatting Concerns:**

1. **Table 4 Precision Labeling Inconsistency​​** The throughput comparison table contains critical errors in precision specification:
Invalid Format Notation:
    + The 1.5B Llama entry erroneously lists "​​BF6​​" (Table 4, row 12) rather than the standard "BF16" format used consistently elsewhere.
    + Impact: This inconsistency undermines cross-model performance comparisons and suggests inadequate quality control in data presentation.

2. **Appendix Omission with Referential Violations​​** The manuscript systematically references non-existent appendix content across multiple sections:

    + Section 5.2: Claims weight decay analysis exists "later in the Appendix"
    + Section 5: Promises hyperparameter details "available in the Appendix"

**Quality:**

3

**Strengths And Weaknesses:**

**Strengths**
1. This paper proposes a pure FP8 LLM training method
2. Table 4 shows that FP8 significantly improves throughput

**Weaknesses**

1. The current evaluation scenario is mainly short text scenarios. It is unclear whether FP8 will cause serious value problems in long text scenarios.
2. The volume of data validated in the paper is insufficient. While the authors argue that this aligns with the optimal cost-performance tradeoff, practical implementations (e.g., Llama and DeepSeek) typically employ pre-training datasets that far exceed this "optimal" threshold. Moreover, as stated in Section 4.3 of [1], a 1B-parameter model with FP8 requires training on approximately ~27T tokens to reliably observe significant degradation caused by quantization.
3. The paper appears structurally incomplete, and crucially, key appendixes are missing.

**Reference**

[1] Sun, Xingwu, et al. "Scaling Laws for Floating Point Quantization Training."

---

> ### Author Rebuttal · Authors · 2025-07-31
>
> We thank the reviewer for their time.
> However, we believe several of the concerns raised do not accurately reflect the contributions of our work.
> We address these points below with clarifications and supporting evidence.
>
> > Short text scenarios
>
> We thank the reviewer for pointing out to larger context length scenarios.
> First, let's note that the sequence length of 4096 adopted in our experiments lies within the standard configurations of current small and large scale language model pre-training setups, such as DeepSeek-v3 [1], OLMo2 7B,13B,32B [2], Qwen2.5 0.5B-72B [3], and SmolLM3 3B [4].
> This moderate context-length pre-training stage usually takes the largest amount of time, mainly because of the prohibitive (quadratic) cost of attention. For instance, Qwen3 models were trained with sequence length 4k for more than 35T tokens before switching to longer values for a few hundred billion tokens. It took only 100B tokens out of 11.3T tokens for SmolLM3 to perform the final long context expansion phase. Finally Llama3.1-405B trained on long sequences for only 0.8T tokens out of nearly 16T tokens in total.
> Hence, speeding up the main moderate-context phase of pretraining has the biggest impact on overall training efficiency.
>
> Nonetheless, long context training after the main phase can still be relatively expensive because iterations become considerably slower.
> Our method benefits from longer contexts, because we perform attention computations in FP8.
> Indeed, attention tends to dominate the computational cost of the forward-backward pass **as sequence length increases**, which intuitively **yields even larger gaps compared to previous FP8 approaches**.
>
> **We demonstrate this trend at 8B model scale** by benchmarking and report throughput values (in tokens/second/gpu) results in the following table:
>
> | Sequence Length | LLaMA BF16 | FOG FP8DPA | Smooth-SwiGLU FP8 |
> | --- | --- | --- | --- |
> | 4K  (TP=1) | 9.48K | 13.52K (**+42.6%**) | 13.13K (**+38.5%**) |
> | 8K  (TP=1) | 9.08K | 13.03K (**+43.5%**) | Out Of Memory |
> | 8K  (TP=2) | 7.49k | 10.42K (**+39.1%**) | 10.05K (**+34.2%**) |
> | 16K (TP=2) | 6.81k | 9.45K (**+38.8%**) | 8.93K (**+31.1%**) |
>
> A batch size of 1024 was used for all experiments, providing even better efficiency gains than the ones reported in the paper (+40.2% --> +42.6%).
> Actually, when GBS increases, computation time takes over communication.
> Clearly, the gap between the FP8-only approach (Smooth-SwiGLU) and our FP8DPA setting becomes larger as context increases.
> To be more specific, we do not claim higher accelerations with higher context lengths because such settings generally require more model sharding, resulting in lower boosts. But we do show intuitively and empirically that the gap with existing FP8 methods increases with the context length: which is a strength that we'll make sure to highlight in the next revision.
>
> Finally, we performed an **additional experiment** and ablated Llama BF16 and FOG-max FP8DPA 390M-sized models, for 50B tokens **with a much longer sequence length of 16k**, using the same configuration as specified in our paper at this scale (Table 1 of the Appendix).
> The results were consistent with those reported at 4K sequence length: stable loss progression and equivalent final performance. Therefore there are no *"serious value problems"* with longer sequence length training, but instead the opposite of even stronger efficiency gains.
>
> > The volume of data validated in the paper is insufficient
>
> We would like to present the following 3 arguments to answer this point:
> - **Practical implementations** of state-of-the-art models such as DeepSeek-v3-671B scale data to a point that corresponds to roughly 1.2 x the compute-optimal data budget taken from the chinchilla rule of thumb [5]. The factor is around 11x for Llama-3.1-70B and even less for Llama-3.1-405B. Therefore, we still do believe that scaling training of a 1.5B model to 420 billion tokens, which is 14x the compute-optimal, can be considered as a long data regime at this scale. Especially, given the consistency in performance and stability we've demonstrated when scaling model size.
> - One of this paper's novelties, is the use of kurtosis to monitor and potentially predict future divergences. In Figure 4, we have provided **low kurtosis guarantees** for much longer than the above mentioned practical chinchilla factors, as well as a **sublinear progression** further strengthening the stable data scaling argument.
> - Judging from most of the practical implementations, scaling the training of a 1B model to 27T tokens is almost never done. Further, distillation seems to be the go-to recipe in general to obtain cheap but strong small language models (cf. Gemma3, Falcon3).
>
> > The paper appears structurally incomplete, and crucially, key appendixes are missing
>
> Thank you for your comment.
> We would have appreciated more specificity and clarity regarding the "structural incompleteness". There is a misunderstanding, as our appendix was already provided in the standard supplementary material submission as instructed by NeurIPS, and was always available.
> Regarding the reviewer claims about "missing appendices":
> - "Section 5.2: Claims weight decay analysis exists 'later in the Appendix'".
>     - We kindly redirect the reviewer to **lines 41-46 and validation experiments represented by Table 5** of the supplemental materials uploaded in addition of the main manuscript, where such analysis can be found.
> - "Section 5: Promises hyperparameter details 'available in the Appendix'".
>     - These details are described in **lines 1-6 and clearly represented in Table 1 of the Appendix**, present in the supplemental materials uploaded in addition of the main manuscript.
>
> > Table 4 Precision Labeling Inconsistency
>
> We appreciate the reviewer for pointing out the "BF6" typo in Table 4. We'll make sure to correct it in the next revision.
> However, although we understand the reviewer might have been influenced by the misbelief of "missing appendix", we'd still like to raise our concerns about overemphasising over a clear typo. We'd like to point the reviewer to the comment made by reviewer 5MCk: "This paper is well written, logically clear, and fully expresses the motivation and methods".
>
> We thank reviewer r2ET for the feedback.
> As we addressed all of the concerns, we hope it is now clearer that the paper is technically sound and that no parts are missing. We kindly ask the reviewer to adjust the scores accordingly.
>
> [1]: DeepSeek-AI, et al., 2025, DeepSeek-V3 Technical Report.
> [2]: Team OLMo, et al., 2025, 2 OLMo 2 Furious.
> [3]: Qwen, et al., 2025, Qwen2.5 Technical Report.
> [4]: Elie Bakouch, et al., 2025, SmolLM3: smol, multilingual, long-context reasoner.
> [5]: Jordan Hoffmann, et al., 2022, Training Compute-Optimal Large Language Models, in Proceedings of the 36th International Conference on Neural Information Processing Systems.

---

### Official Review · Reviewer_CRBu · 2025-07-03

**Clarity:** 2
**Significance:** 3
**Originality:** 4
**Rating:** 5
**Confidence:** 2

**Summary:**

This paper introduces FOG (Fast and Outlier-Guarded), a novel set of LLM architectures specifically designed to enable stable FP8 training with all GEMMs, including attention computations (FP8DPA). The authors address a critical limitation in current FP8 training approaches that either use suboptimal fine-grained scaling or fall back to higher precision in sensitive components. The key innovation is architectural modifications that reduce activation outliers, enabling the use of simple delayed scaling strategies while achieving throughput improvements of up to 40%.

**Questions:**

See weaknesses

**Ethical Concerns:**

["NO or VERY MINOR ethics concerns only"]

**Final Justification:**

During the discussions with the authors, the problem mentioned in the initial reviews has been mainly resolved, with confirmation of updating in the next version. I believe this work offers a comprehensive framework for FP8DPA training. Based on this information, I have raised my score.

**Limitations:**

See weaknesses

**Quality:**

3

**Strengths And Weaknesses:**

## Strengths

### 1. **Significant Technical Achievement**
This is the first work to successfully demonstrate stable FP8 computation in attention mechanisms at scale. The ability to use FP8 for all GEMMs within transformer blocks represents a meaningful step toward fully low-precision LLM training.

### 2. **Comprehensive Empirical Validation**
- Extensive experiments across multiple model scales (390M, 1.5B, 8B parameters)
- Long-term training validation (up to 420B tokens, 14× Chinchilla optimal)
- Thorough comparison with existing architectures shows consistent divergence in competitors
- Downstream performance matching or exceeding BF16 baselines

### 3. **Practical Impact**
- Substantial throughput gains (35-40%) with maintained model quality
- Uses simple delayed scaling rather than complex fine-grained approaches
- Provides actionable architectural guidelines for the community

### 4. **Novel Diagnostic Tool**
The use of kurtosis as an early warning system for FP8 training divergence is innovative and potentially valuable for future research. The observation that kurtosis exhibits abnormal behavior well before loss divergence could be broadly applicable.

### 5. **Thorough Ablation Studies**
The step-by-step transition from OP to FOG architectures clearly identifies the critical components (frozen QK gains, post-normalization) responsible for stability.

## Weaknesses

### 1. **Definition of xIELU**
While xIELU is a new activation function that is not commonly used, I believe a definition is necessary in this paper for readers to understand it.

### 2.  **Further Explanation of the FP16 baseline**
The proposed method entails several architectural changes compared to the existing LLM architecture in LLaMA3. Therefore, I would expect the authors to provide more comparisons between the two architectures in the experiments. Moreover, I still do not understand why it can match the original FP16 LLaMA3 with pre-norm and SwiGLU.

### 3.  **Loss Drop of Llama3 in Figure 7**
From Figure 7, the llama3 baseline has a much sharper loss drop after 100B tokens. Can the authors provide further explanation for this phenomenon?  Moreover, why is this drop not a good feature, but some other feature with worse smoothness?

### 4. **Lack of ablation**
I would expect some ablation for llama3 to the new architecture in FP16, demonstrating how the architecture itself affects performance.

---

> ### Author Rebuttal · Authors · 2025-07-31
>
> We thank the reviewer for their time and thoughtful feedback.
> We are glad they found many aspects of the work promising.
> Below, we address the questions and concerns raised.
>
> > Definition of xIELU
>
> Although we provide the reference work [1] that introduces xIELU in line 186 of the submitted manuscript, we agree that this non-linearity has been proposed relatively recently. And adding its definition to the manuscript would give readers a clearer picture of FOG-MAX variant. We thank the reviewer for pointing this out and will provide a clear definition in the appendix upon acceptance.
> As we mention xIELU, let's note that this choice was deliberate. It's an inherently quadratic function and thus is an outlier-amplifier. Therefore, choosing an architecture with xIELU while demonstrating that our method still remains very stable highlights FOG's significant robustness to outliers, pointing to the limits of the excellent former work [2], as discussed in paragraph: L183-L191.
> We believe that the FOG's robustness when using quadratic activation functions is an important strength of our work that we should make clearer in the next revision.
>
> > Loss Drop of Llama3 in Figure 7
>
> The loss drop pointed out in Figure 7 is expected, and matches the LR cooldown during the last phase of the widely adopted WSD LR schedule.
> This phenomenon is positive and allows matching cosine LR schedule's final performance [3]. Note that the drop is consistently observed for all architectures reaching the final LR cooldown phase, such as around 40BT in Figure 2; around 40BT for the 390M model and 100BT for the 1.5B model in Figure 6.
> For the long FOG-Max-1.5B run trained on over 420BTs, LR cooldown hasn't been applied yet as the only goal was showcasing stability. To address the reviewer's concern, we've pursued this run by **conducting an extra 25BTs of LR cooldown**, which resulted in an expected loss drop reaching a final value of **2.27**, improving upon Llama-1.5B final loss in Figure 7.
>
> > Further Explanation of the FP16 baseline + Lack of ablation
>
> We first clarify that BF16 is the standard half-precision format used for training [4]. Although model weights can be stored in FP16, GEMMs are done in BF16 for more dynamic range. In the following we assume the reviewer refers to this widely adopted setting, that we call BF16 training.
>
> *"I would expect the authors to provide more comparisons between the two architectures in the experiments"*:
> We kindly refer the reviewer to Table 4 of the supplemental materials, where we report a wide variety of **downstream performance** results following extensive experiments to compare FOG variants against the Llama BF16 baseline. We note that for each task and each FOG variant, downstream performances are provided for both BF16 training and FP8DPA training. We also point to Figure 4 where **upstream performance (loss)** aligns with task performances, and to Table 4 where we report **throughput comparisons**.
>
> *"I still do not understand why it can match the original FP16 LLaMA3 with pre-norm and SwiGLU"*:
> Table 4 shows that all BF16 FOG-variants exhibit similar performances to original BF16 Llama3. Indeed, FOG is designed to preserve both stability and performance when reducing GEMMs precision to FP8, even when FP8 computation of the dot product attention (FP8DPA) is carried out. This is highlighted by the scores reached with FP8DPA training as well as the equivalent final losses. We explain this "equivalent performance with both BF16 and FP8DPA" by **the robustness of the architecture** as well as the choices of **expressive activation functions** and beneficial **QK-regularization**.
> Further, we note some consistent trends such as FOG-max slightly outperforming other architectures, including baseline. In [1], it has been shown that xIELU outperforms SwiGLU thanks to its linearly increasing gradient for positive inputs and its trainable gradient for negative inputs. QK-RMSNorm and QK-Tanh play a role of entropy regularizers, preventing spiky attention softmax outputs, crucially contributing to FOG's performance and stability. In the following table where we introduce two additional experiments, we re-confirm the stabilizing effect of QK regularization under FP8DPA regime, by the delayed divergence. The starting point of the architecture ablated below is now Llama and QK is introduced after Post-LN.
>
> | Size=390M (FP8DPA)     | Data Budget | Normalization | QK-RMSNorm | Activation Function   | Observation                    |
> |-------------------------|-------------|----------------|---------|----------------------|----------------------------------|
> | Llama3           | 50B tokens  | Pre-LN         | No      | SwiGLU                      | Early divergence (Fig-6)         |
> | Llama3 w/ Post-LN        | 50B tokens  | Post-LN        | No      | SwiGLU              | Early divergence (new)           |
> | oLMo                    | 50B tokens  | Post-LN        | Yes     | SwiGLU               | Mid-training divergence (Fig-6)  |
> | FOG-SwiGLU-1.5B**              | 125B tokens | Post-LN        | Yes *   | SwiGLU        | Convergence, equivalent to baseline (new) |
>
> *Frozen QK-norm gains.
> **Note that the FOG-SwiGLU variant test scales up model size and data budget compared to the other 390M architectures.
>
> To conclude, our work shows that FP8 precision is sufficient to reach the long standing half-precision performances **without trade-offs**, while providing **flexible architectural choices** (QK-regularizers, sub-linear/quadratic/gated activations) with a training **acceleration that scales better with long context** than all previous approaches, thanks to the FP8 attention computation.
>
>
> [1]: Allen Hao Huang, Imanol Schlag, 2025, Deriving Activation Functions Using Integration.
> [2]: Maxim Fishman, Brian Chmiel, Ron Banner, Daniel Soudry, 2025, Scaling FP8 training to trillion-token LLMs, in The Thirteenth International Conference on Learning Representations.
> [3]: Alexander Hägele, Elie Bakouch, Atli Kosson, Loubna Benallal, Leandro Von Werra, Martin Jaggi, 2024, Scaling Laws and Compute-Optimal Training Beyond Fixed Training Durations, in The Thirty-eighth Annual Conference on Neural Information Processing Systems.
> [4]: Dhiraj Kalamkar, et al., 2019, A Study of BFLOAT16 for Deep Learning Training.

---

> ### Comment · Reviewer_CRBu · 2025-07-31
> **Response from Reviewer CRBu**
>
> Thank you very much for the rebuttal!
>
> First, based on your responses to Q1 and Q4, I would appreciate it if we could both clarify the definition of xIELU and include the SwiGLU variants in the main experiments, as it can also work under the FP8DPA regime. I understand the xIELU may be more powerful than SwiGLU. However, it is not widely adopted today, so it remains to be seen whether it can scale to larger models. Therefore, I think further results with SwiGLU + FP8DPA with other designs such as QK Norm (becoming common these days) will enhance the quality of this work. People can try your method with fewer component replacements.
>
> Second, based on your explanation, I understand the meaning of Figure 7. However, it is hard for the audience to distinguish between these two training curves (one with annealing and one without). Therefore, I recommend that the authors use the same training configuration in the Figure for the next version.
>
> Moreover, I was wondering whether the authors could provide some comparison or discussion with DeepGEMM, proposed by Deepseek, and MXFP8 on NVIDIA GPUs.  I was wondering whether these designs can lead to more stable FP8 training and enable FP8DPA.
>
> Additionally, for the authors' response to other reviewers, I suggest focusing on additional experiments that optimize FP8 states. I was wondering whether the authors could provide more details on this part, including both the precision for momentum and variants, and the througputs after applying FP8 optimization states.
>
> Thank you very much!

---

> > ### Author Response · Authors · 2025-08-03
> >
> > We thank the reviewer for their reactivity, constructive feedback and recommendations!
> >
> > > “ I would appreciate it if we could both clarify the definition of xIELU and include the SwiGLU variants in the main experiments “
> >
> > We fully agree with both suggestions, especially the inclusion of the ablation on FOG-SwiGLU variant in the main paper.
> > Indeed, such an addition generalizes FOG architecture's robustness to Gated activations (commonly used) and allows less modifications from mainstream settings as pointed by the reviewer .
> >
> > > However, it is hard for the audience to distinguish between these two training curves (one with annealing and one without).
> >
> > Totally. As we already resumed the long-data-regime experiment with FOG-Max by conducting a LR annealing on 25BTs, we will update Figure 7 with this additional cooldown as well as its legend with a brief explanation.
> >
> > > “ I was wondering whether the authors could provide some comparison or discussion with DeepGEMM, proposed by Deepseek, and MXFP8 on NVIDIA GPUs “
> >
> > This is indeed an important question.
> > - First, regarding **DeepGEMM** proposed by DeepSeek, its main idea is opting for a fine-grained scaling strategy instead of standard tensorwise scaling approaches such as DelayedScaling (**D.S**), which we employ. DeepGemm also comes with a few other ideas such as the "promotion" technique (two-stage accumulation). Recently, inspired by DeepGEMM, Nvidia has officially released **Blockwise Scaling** (B.S) support for Hopper GPUs used in this work, optimized for their FP8 tensor cores. Intuitively, this fine-grained scaling strategy should be "safer". It's better suited to accommodate the potential block-level variabilities within the same tensor. Again, intuitively and despite optimizations, it leads to considerably lower boosts due to the need of computing and applying a larger number of scaling factors. For instance, it would necessitate more than 16K factors instead of 1, in the case of a 4096x16834x1 GEMM with block-size=64x64. Now empirically, **we have benchmarked this blockwise scaling approach** on Nvidia's GH200 at 8B scale, confirming the lower throughput gains it provides, as shown by the following table. We think these results likely generalize to DeepGEMM.
> >
> > | Architecture | Precision | FP8 recipe | Throughput (tokens/sec/gpu) |
> > | :- | :-: | :-: | :-: |
> > | Llama | BF16 | N/A | 9.48k |
> > | Llama | FP8 | Blockwise* | 11.18k (+17.9%) |
> > | OP | FP8 | DelayedScaling | 12.14k (+28.1%) |
> > | Llama+SmoothSwiGLU | FP8 | DelayedScaling | 13.1k (+38.2%)  |
> > | FOG-flash | FP8DPA | DelayedScaling | 13.52k (+42.6%) |
> >
> > \* NVIDIA's Transformer Engine implementation.
> >
> > - Second, MXFP8 is a hardware-level implementation of blockwise scaling with few optimizations, where the chip is natively handling the scaling. However, it comes with the following limitations:
> >     - Block size can only be 32x32 with MXFP8, which is a relatively small size: likely good for stability but leads to slower GEMMs. With blockwise scaling on Hopper, we can configure the block size to be larger, but we don't have hardware support for micro-scaling as in blackwell.
> >     - We have **no guarantees of MXFP8 stability under FP8DPA regime with standard architectures** (e.g. Llama). Unfortunately, we couldn't provide Blackwell GPUs so that we can test its potential to ensure stable FP8DPA training with standard architectures. We note that DeepSeek-v3 training opted for higher precision attention computation. Now, assuming it does, the scaling strategy itself and the fact that it resembles blockwise scaling, which is shown to be much slower, is unlikely to reach or exceed simple DelayedScaling boosts.
> >     - In any case, many future trainings will still use Hopper GPUs or similar chips that do not support MX natively. We have shown that Blockwise scaling, supported by Hopper/Intel Gaudi/AMD Ml300, is slower than D.S. This means that even if MXFP8 surprisingly reaches FP8 D.S boosts with new hardware, **our approach is valuable for the substantial installed base of non-MX hardware** (s.a Nvidia Ada, Nvidia Hopper).
> >
> > > “ I suggest focusing on additional experiments that optimize FP8 states “
> >
> > We kindly point to our rebuttal with Reviewer nVZB where we describe **an additional experiment that showcases stable training** under FP8DPA training regime and **FP8 moments** {FP8 moments, BF16 gradients, FP16 weights}, further saving memory compared to our initial setting with FP16 optimizer states. However, these choices impact memory usage--not extremely as we already shard the optimizer states with Zero1--with no effect on throughput. Indeed, throughput is mainly bounded by the forward-backward pass computation/communication.
> > We will make sure to include this FP8-states ablation in the next revision.
> >
> > We hope these clarifications address your concerns and would be grateful if you could reconsider the evaluation in light of these responses.

---

> > > ### Comment · Reviewer_CRBu · 2025-08-05
> > > **Response from Reviewer CRBu**
> > >
> > > Dear authors,
> > >
> > > Thank you for your continuing response! I think these discussions are very useful. Based on this information, I have raised my score to 5 (In case OpenReview hides the modified score).
> > >
> > > It will be very interesting to further explore this direction! Especially the hardware quantization in new hardware and using FP8 in more cases.

---

> > > > ### Author Response · Authors · 2025-08-05
> > > >
> > > > Dear Reviewer CRBu,
> > > > Thank you for your thoughtful engagement and re-evaluation!
> > > > We're also excited about further exploring this direction with upcoming hardware capabilities.

---

### Note · Authors · 2025-08-13

Dear AC,

We thank you and the reviewers for your efforts in evaluating our work!

We enjoyed the discussion period, which led to improved ratings from all reviewers except reviewer 5MCk.
We consider this final remarks section important for a few last clarifications.

Let us start with reviewers CRBu and r2ET, who engaged most with the rebuttal.
We are pleased that all their concerns and misunderstandings were resolved, and that new strengths of our work were recognized (e.g., long-context training), leading them to lean toward clear acceptance.

Reviewer nVZB provided a thorough initial review, which we addressed with explanations and new experiments:
 - **Generalization to MoEs**, though not initially the main focus (new experiments at 1.8B and 41.5B scales).
 - **Maintained stability and performance with FP8 optimizer states**, further reducing memory use (0.4B scale).
 - **Generalization to gated activations**, challenging claims of the best prior method, Smooth-SwiGLU (1.5B scale).
 - **Even greater acceleration gains** over prior methods in **long-context** scenarios (exhaustive benchmarking at 8B scale).

Along with reviewer CRBu, they valued the novel diagnosis and early warning metric: **kurtosis**.
Reviewer nVZB announced raising their rating to borderline accept, although **no remaining concerns were noted**.

Now, most importantly, our rebuttal to reviewer 5MCk showed that **3 of 4 claimed weaknesses were incorrect**—the missing explanations, ablations, and generalizations were already present in the submission. We clarified these points, and the reviewer acknowledged that the rebuttal improved their understanding.
The remaining discussion focused on the motivation (reducing outliers), which the reviewer considered heuristic.
We explained that it is supported both inductively (kurtosis analysis, stability ablations) and deductively (Introduction), and we provided two additional **detailed theoretical arguments with new empirical confirmation** (weight analysis of FOG and Llama).
We also summarized broader feedback from other reviewers to aid their re-evaluation.

However, we received no final response, and the borderline reject recommendation remains.
We therefore kindly ask the AC to seek clarification on this matter.


Again, we thank all parties for their time and efforts, and hope this work will benefit many researchers, as reviewer r2ET also trusted.


Best regards,
Authors of submission n°22812

---

### Decision · Program_Chairs · 2025-09-17

**Decision:**

Accept (poster)

**Comment:**

This paper introduces a new architecture that aims to eliminate the outlier and enable the fully FP8 training of the LLM, and achieve the BF16 performance at scale. The author conducted medium-scale experiment with downstream evaluation to show the effectiveness of the proposed architecture. Most reviews agree on the significance of this contribution. However, concerns are raised regarding the insufficient ablation studies, and deeper understandings on why this eliminates the outlier. Nevertheless, this work is a nice contribution to the community and enable future research about low-precision training.